# Continuous Speculative Decoding for Autoregressive Image Generation

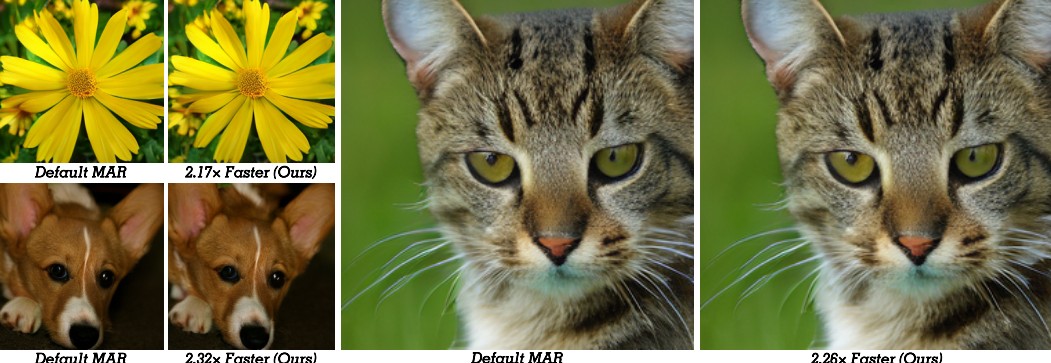

Figure 1: Continuous speculative decoding accelerates the inference speed while maintaining the original generation quality. For each panel, left: image generated by MAR; right: image generated by MAR with continuous speculative decoding. Speed-up ratio of each single image is presented.

## Abstract

Continuous visual autoregressive (AR) models have demonstrated promising performance in image generation. However, the heavy autoregressive inference burden imposes significant overhead. In Large Language Models (LLMs), speculative decoding has effectively accelerated discrete autoregressive inference. However, the absence of an analogous theory for continuous distributions precludes its use in accelerating continuous AR models. To fill this gap, this work presents continuous speculative decoding, and addresses challenges from: 1) low acceptance rate, caused by inconsistent output distribution between target and draft models, and 2) modified distribution without analytic expression, caused by complex integral. To address challenge 1), we introduce an approximate acceptance criterion to address the inefficiency in sampling. Furthermore, we propose denoising trajectory alignment and token pre-filling strategies. To address challenge 2), we introduce acceptance-rejection sampling algorithm with an appropriate upper bound, thereby avoiding explicitly calculating the integral. Furthermore, our denoising trajectory alignment is also reused in acceptance-rejection sampling, effectively avoiding repetitive diffusion model inference. Extensive experiments demonstrate that our proposed continuous speculative decoding achieves over $2\times$ speedup on off-the-shelf models, while maintaining the generation quality.

## 1 Introduction

Autoregressive (AR) models have demonstrated significant potential and achieved competitive performance in image generation tasks (Van Den Oord et al., 2016; Van den Oord et al., 2016; Esser et al., 2021; Yu et al., 2023a; Tian et al., 2025). These models predict next token sequentially based on previously generated tokens, outputting discrete categorical distributions. Typically, the input image is mapped from the pixel space to tokens through vector quantization (VQ), after which the AR model generates images by predicting discrete distributions for subsequent tokens. While this process shows substantial promise in image generation, VQ can lead to training instability and might

Figure 2: Comparison between discrete and continuous speculative decoding. Discrete situation offers the convenience of directly computing probabilities and simply sampling from modified distributions. In contrast, continuous situation faces challenges in the inconsistency of output distributions, leading to low acceptance criterion as well as low acceptance rate, and the modified distributions without analytic expression, caused by complex integral.

not adequately capture the nuanced image details (Yu et al., 2023a; Mentzer et al., 2023). Recently, continuous visual AR models have been proposed to predict visual tokens from continuous probability distributions (Tschannen et al., 2025; Li et al., 2025; Xu et al., 2024; Sucheng et al., 2025). In this framework, the model predicts next token's continuous distribution based on prior outputs, often implemented via denoising processes (Ho et al., 2020; Nichol & Dhariwal, 2021). This approach overcomes the limitations of VQ, offering a promising solution for autoregressive image generation.

However, continuous visual AR models suffer from slow inference speed due to the sequential decoding process, a limitation shared with LLMs. LLMs commonly employ speculative decoding (Leviathan et al., 2023; Chen et al., 2023) to accelerate autoregressive inference procedure. This algorithm employs a draft-and-verification mechanism, where a smaller draft model generates several draft tokens, and then a more accurate yet often larger target model verifies them. Inspired by this, previous works have investigated the efficacy of speculative decoding for accelerating discrete visual AR models (Jang et al., 2024; Teng et al., 2024). However, the related methods are formulated for discrete output distributions, whereas a continuous formulation remains undeveloped.

To bridge this gap, we present **Continuous Speculative Decoding**, designed to accelerate the inference of continuous visual AR models. However, as illustrated in Figure 2, the following two challenges can be identified while developing this method: a) **Inconsistent output distribution.** There is significant inconsistency in diffusion and autoregressive procedures, resulting in distinct draft and target output distributions. This leads to a very low acceptance criterion, thus lowering the acceptance rate. b) **Modified distribution without analytic expression.** When an output token is rejected, a new token will be drawn from the modified distribution. In continuous distributions, there is no analytic expression for this due to the complex integral for normalizing the probability distribution, making it impossible to directly sample from this distribution.

To address the aforementioned challenges, this work proposes a series of technical optimizations. **First**, to improve the acceptance rate, we introduce an approximate acceptance criterion to address the inefficiency in sampling. Furthermore, we develop denoising trajectory alignment method based on the proposed theorem of *proximity in the reparameterization*. This method aims to reduce the distance between two output distributions, thereby resolving the inconsistency in the diffusion procedure. Additionally, we introduce a token pre-filling strategy to improve the early low acceptance rate. This strategy leverages the principle that the output of AR models depends on previous states, addressing inconsistencies within the autoregressive procedure. **Second**, to effectively sample from the complex modified distribution, we introduce the acceptance-rejection sampling algorithm (Casella et al., 2004) by deriving a proper upper bound, which helps to eliminate the need for computing complex integral. Besides, to avoid repetitive use of diffusion model for inference during acceptance-rejection sampling, we derive an easy-to-compute rejection threshold by reusing the denoising trajectory alignment.

Our continuous speculative decoding can be integrated seamlessly into many of existing models, as shown in Figure 1. We validate the effectiveness of our algorithm on three continuous visual AR models (Li et al., 2025; Sucheng et al., 2025; Wu et al., 2025) at two resolutions (256 & 512) through qualitative and quantitative evaluations. Specifically, we measure wall-time improvements and report image generation performance using Fréchet Inception Distance (FID) (Heusel et al., 2017)

and Inception Score (IS) (Salimans et al., 2016). Extensive experiments show that our algorithm achieves over $2\times$ inference speedup while maintaining generation quality.

Our contributions can be summarized as follows:

- We are the first to propose continuous speculative decoding, bridging the gap of speculative decoding to continuous distributions and enabling substantial acceleration of continuous visual AR models.

- We address the distinct problem of low acceptance rate in continuous speculative decoding with an approximated acceptance criterion by introducing two novel techniques: denoising trajectory alignment, which reduces the draft-target distribution distance, and token pre-filling, which mitigates the early low acceptance rate problem.

- We solve the hard-to-sample problem of modified distribution via acceptance-rejection sampling with a proper upper bound to avoid the complex integral and an easy-to-compute rejection threshold with denoising trajectory alignment to avoid repetitive model inference.

- We seamlessly integrate our algorithm into three existing continuous visual AR models without extra training or architectural changes. Extensive experiments show that it achieves over $2\times$ inference speedup while maintaining generation quality.

## 2 RELATED WORK

### 2.1 AUTOREGRESSIVE IMAGE GENERATION

Autoregressive (AR) models are widely used in image generation. Early works perform generation at pixel level using CNN (Van den Oord et al., 2016; Chen et al., 2018), RNN (Van Den Oord et al., 2016) and Transformer (Parmar et al., 2018; Chen et al., 2020). VAR (Tian et al., 2025) modifies the autoregressive paradigm into next-scale prediction to gradually increase the scale of predictions. Similar to the autoregressive language model, AR image generation through discrete token prediction is scalable to text-conditioned image generation (Liu et al., 2024; Sun et al., 2024; Yu et al., 2023b). However, training discrete image tokenizer is difficult, and its ability to convey detailed visuals is still questionable (Yu et al., 2023a; Mentzer et al., 2023). GIVT (Tschannen et al., 2025) represents continuous tokens via Gaussian mixture models. MAR (Li et al., 2025) and DisCo-Diff (Xu et al., 2024) generate tokens via diffusion process (Ho et al., 2020) conditioned by the autoregressive model. HART (Tang et al., 2024) employs discrete and continuous tokenizer to generate images, with classification for discrete tokens and denoising for the residual between primitive visual tokens and discrete tokens. xAR (Sucheng et al., 2025) extends the conception of token and reformulates discrete token classification as continuous entity regression. However, autoregressive models suffer from heavy inference overhead. The inference speed is slowed down by step-by-step generation.

### 2.2 SPECULATIVE DECODING

Speculative decoding (Leviathan et al., 2023; Chen et al., 2023) achieves lossless acceleration by verifying the draft model with the target model. Following this, previous works mainly focus on reducing draft model overhead and strengthening the consistency between the draft and target models. SpecInfer (Miao et al., 2023) employs multiple small draft models and aggregates their predictions into a tree structure to be verified through tree-based parallel decoding. Eagle (Li et al., 2024b;c) improves the draft accuracy through the prediction at the feature level instead of the token level to tackle the feature uncertainty problem. Jacobi iteration is also employed to reduce inference overhead in the decoding process (Santilli et al., 2023; Zhao et al., 2024b;a; Kou et al., 2024). Online Speculative Decoding (Liu et al., 2023) and DistillSpec (Zhou et al., 2023) align the output from the draft model with the target model with more training.

Recent works are now using speculative decoding to improve the efficiency of autoregressive image generation. SJD (Teng et al., 2024) improves the Jacobi iteration process by adding speculative decoding while keeping the variety of image generation. LANTERN series (Jang et al., 2024; Park et al., 2025) looks at distribution ambiguity and uses relaxation to add more flexible candidate tokens, maintaining high image quality. These studies have greatly speeds up the process by reducing

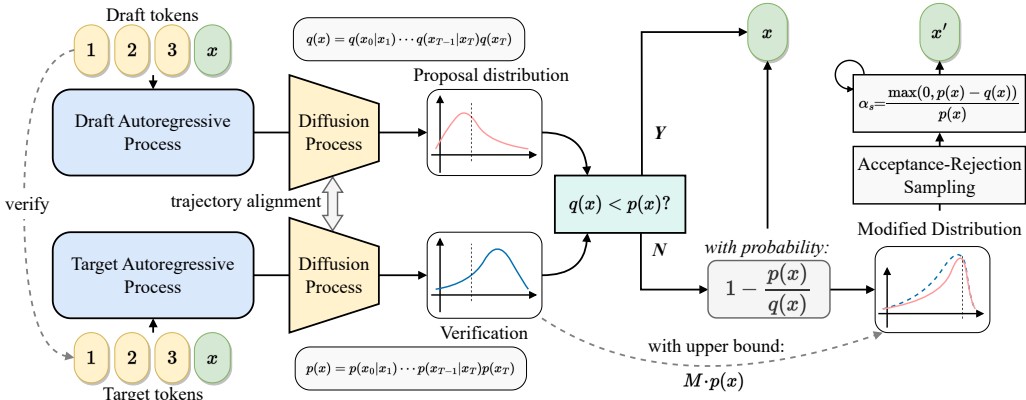

Figure 3: The overview of continuous speculative decoding. The diffusion model component of continuous AR models is leveraged. Tokens $1 \sim 3$ are prefix tokens, and token $x$ is to be verified. In this method, $x$ is accepted if $q(x) < p(x)$. Otherwise, $x$ is rejected with probability $1 - p(x)/q(x)$, followed by sampling $x'$ from the modified distribution via acceptance-rejection sampling.

the inference steps needed for generating visual tokens. However, they are only applicable to discrete space. By contrast, our work extends speculative decoding to continuous AR models.

## 3 METHODOLOGY

### 3.1 FROM DISCRETE TO CONTINUOUS SPECULATIVE DECODING

We first introduce the discrete form of speculative decoding. It utilizes a draft model $M_q$ with output distribution $q(x)$ and a target model $M_p$ with $p(x)$. $M_q$ generates a sequence of draft tokens $x_{i:j} = \{x_i, \ldots, x_j\}$ where $x \sim q(x)$, which are then verified by $M_p$ in parallel. $x$ is accepted if the acceptance criterion $p(x)/q(x) > 1$; otherwise, it is rejected with probability $1 - p(x)/q(x)$ and resampled from a modified distribution $p'(x) = norm(max(0, p(x) - q(x))) = \frac{max(0, p(x) - q(x))}{\sum_{x'} max(0, p(x') - q(x'))}$.

Then, we discuss continuous speculative decoding (see Figure 3). In continuous visual AR models (Li et al., 2025), the output distribution is typically modeled via diffusion models, namely:

$$p(x_{N,0:T}, |x_{1:N-1}) = p(x_{N,T}) \prod_{t=1}^{T} p(x_{N,t-1}|x_{N,t}, x_{1:N-1}), \tag{1}$$

where $N$ represents $N$-th autoregressive step. $t \in [0, T]$ is the diffusion timestep. $x_{N,t}$ represents the state of $x$ at AR step $N$ and diffusion timestep $t$. $x_{1:N-1} = \{x_1, x_2, \ldots, x_{N-1}\}$ denotes the variables before $N$. For simplicity, we omit $N$ and $x_{1:N-1}$ and let $Y = x_{0:T}$ to obtain $p(Y) = p(x_T) \prod_{t=1}^{T} p(x_{t-1}|x_t)$.

**Acceptance Criterion** The acceptance criterion is defined as the ratio of its probability under target distribution to the one under draft distribution, that is, $p(x_{0:T})/q(x_{0:T})$. In discrete form, the probability can be directly obtained. But in continuous form, the probability is usually obtained via diffusion process (Ho et al., 2020; Song et al., 2020). Specifically, $x_{0:T}$ is sampled from draft model via reverse diffusion (denoising) process. So the ratio is given by:

$$\frac{p(Y)}{q(Y)} = \frac{p(x_T) \prod_{t=1}^{T} p(x_{t-1}|x_t)}{q(x_T) \prod_{t=1}^{T} q(x_{t-1}|x_t)}, \tag{2}$$

where $x_T$ is a Gaussian noise and $p(x_{t-1}|x_t)$ is approximated as a Gaussian distribution through a neural network $\theta$ (Nichol & Dhariwal, 2021), and $\mu_\theta(x_t, t)$ and $\Sigma_\theta(x_t, t)$ are mean and variance predicted by $\theta$, that is:

$$p(x_{t-1}|x_t) = \mathcal{N}(x_{t-1}; \mu_\theta(x_t, t), \Sigma_\theta(x_t, t)). \tag{3}$$

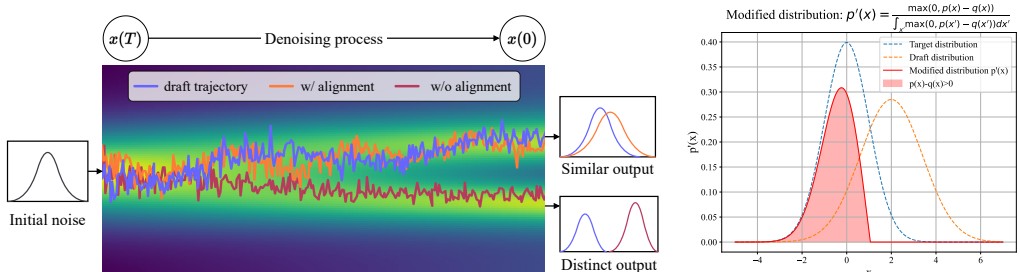

Figure 4: Illustration of denoising trajectory alignment. The denoising process maps the noise distribution to data distribution through gradual denoising. These denoising steps form a trajectory. The aligned trajectory (orange curve) leads to a similar output distribution, while the unaligned one (red curve) produces a far-away one, obtaining low $p(Y_p)/q(Y_q)$.

Figure 5: Illustration of distributions. Dashed lines: draft and target distributions. Red area: modified distribution (unnormalized, omitting $Z$ for simplicity), whose integral lack an analytic expression.

However, this approach is impractical because the acceptance rate is low. As discussed in Appendix A.1.1, the draft model's trajectory $x_{0:T}$ inherently diverges from the target model's expected trajectory. In each denoising step, samples drawn from the draft model's distribution $q$ are unlikely to fall near $\mu$ of the target distribution $p$, which results in a low single-step ratio $p/q$. As the multi-step denoising process proceeds, the overall $p(Y)/q(Y)$ becomes extremely small.

Therefore, our work adopts a practical approximation: the ratio of the joint probabilities $p(Y_p)/q(Y_q)$, where both $Y_p$ and $Y_q$ share the same $x_0$. This ratio serves as a surrogate for the shared path ratio. Then, the ratio is calculated through:

$$\frac{p(Y_p)}{q(Y_q)} := \frac{p(x_T^p)\prod_{t=1}^{T} p(x_{t-1}^p|x_t^p)}{q(x_T^q)\prod_{t=1}^{T} q(x_{t-1}^q|x_t^q)}, \quad x_0^p = x_0^q = x_0, \tag{4}$$

**Modified Distribution**  When a draft token is rejected, a new one is sampled from the modified distribution $p'(x)$. $p'(x)$ is derived from the normalization of $max(0, p(x) - q(x))$. Therefore, replacing the summation in the normalization denominator of the discrete form with integral yields the continuous form, namely:

$$p'(Y) = \frac{max(0, p(Y) - q(Y))}{Z}, \text{ where } Z = \int_{Y'} max(0, p(Y') - q(Y'))dY'. \tag{5}$$

However, in practical operation, an extremely low acceptance rate emerges, as shown in Table 4, thereby leading to poor acceleration performance. To address this, we propose the following technical optimizations.

### 3.2  MITIGATING DISTRIBUTION INCONSISTENCY

Under the acceptance criterion given by Equation 4, the acceptance rate $\alpha$ is extremely small (nearly 0%). We attribute this to two sources of inconsistency in continuous visual AR models: **1) inconsistency in diffusion process**, and **2) inconsistency in autoregressive process**.

**Inconsistency in Diffusion Process**  Significant inconsistency exists in the denoising process. As illustrated in Figure 4, draft and target denoising trajectories diverge to different outputs. The output distance between the two distributions is large. Therefore, for a draft output $x_0$, $q(Y_q)$ is large while $p(Y_p)$ is quite small, leading to a low $p(Y_p)/q(Y_q)$, consequently leads to low acceptance rate $\alpha$.

We propose denoising trajectory alignment to enhance the consistency of the output distributions. Note that in Equation 3, $x_{t-1}^p$ is obtained via reparameterization given by $x_{t-1}^p = \sqrt{\Sigma_\theta^p(x_t^p, t)} \cdot \varepsilon_t^p + \mu_\theta^p(x_t^p, t)$, where $\varepsilon_t^p \sim \mathcal{N}(0, \mathrm{I})$ (same for $x_{t-1}^q$). Denoising trajectory alignment can reduce the expected distance between $x_{t-1}^p$ and $x_{t-1}^q$, promised by the following theorem.

**Theorem 1 (Proximity in the Reparameterization)** *Setting $\varepsilon_t^p = \varepsilon_t^q$ in reparameterization reduces the expected distance $\mathbb{E}\left[\left\|x_{t-1}^q - x_{t-1}^p\right\|^2\right]$ between $x_{t-1}^p$ and $x_{t-1}^q$ by $2 \cdot tr\left[\sqrt{\Sigma_t^q \Sigma_t^p}\right]$.*

Detailed proofs can be found in Appendix A. We emphasize that reducing the distance of output samples aims to increase $p(x)$, thereby increasing $p(x)/q(x)$ for higher acceptance rate. Note that:

$$p(x_{t-1}|x_t) = \frac{1}{(\sqrt{2\pi})^n \sqrt{|\Sigma_\theta(x_t,t)|}} \exp\left\{\frac{1}{2}[x_{t-1} - \mu_\theta(x_t,t)]^T \Sigma_\theta^{-1}(x_t,t)[x_{t-1} - \mu_\theta(x_t,t)]\right\}$$

$$= \frac{1}{(\sqrt{2\pi})^n \sqrt{|\Sigma_\theta(x_t,t)|}} \exp\left\{\frac{1}{2}\varepsilon_t^T \varepsilon_t\right\}, \quad x_t \in \{x_t^p, x_t^q\}, \varepsilon_t \in \{\varepsilon_t^p, \varepsilon_t^q\}. \tag{6}$$

The ratio $p(x_{t-1}^p|x_t^p)/q(x_{t-1}^q|x_t^q)$ given $\varepsilon_t = \varepsilon_t^p = \varepsilon_t^q$ is:

$$\frac{p(x_{t-1}^p|x_t^p)}{q(x_{t-1}^q|x_t^q)} = \frac{\frac{1}{(\sqrt{2\pi})^n \sqrt{|\Sigma_t^p|}} \exp\left\{\frac{1}{2}\varepsilon_t^T \varepsilon_t\right\}}{\frac{1}{(\sqrt{2\pi})^n \sqrt{|\Sigma_t^q|}} \exp\left\{\frac{1}{2}\varepsilon_t^T \varepsilon_t\right\}} = \frac{\sqrt{|\Sigma_t^q|}}{\sqrt{|\Sigma_t^p|}}. \tag{7}$$

For simplicity, we define:

$$\Sigma = \prod_{t=2}^{T} \sqrt{|\Sigma_t^q|} \Big/ \prod_{t=2}^{T} \sqrt{|\Sigma_t^p|}. \tag{8}$$

Substituting $\Sigma$ into Equation 4 makes (assuming $p(x_T^p) = q(x_T^q)$):

$$\frac{p(Y_p)}{q(Y_q)} = \frac{p(x_T^p)p(x_0|x_1^p) \prod_{t=2}^{T} p(x_{t-1}^p|x_t^p)}{q(x_T^q)q(x_0|x_1^q) \prod_{t=2}^{T} q(x_{t-1}^q|x_t^q)} = \frac{p_\theta(x_0|x_1^p)}{q_\theta(x_0|x_1^q)} \cdot \Sigma. \tag{9}$$

Therefore, at each step, employing the same $\varepsilon_t$ reduces the expected distance by $2 \cdot tr\left[\Sigma_t^q \Sigma_t^p\right]$, which enables the two models to generate closer samples and finally increases the acceptance rate.

**Inconsistency in Autoregressive Process** Inconsistency also arises in autoregressive steps. Figure 9 shows that acceptance rate $\alpha$ is very low (5%) at the initial AR steps, and it increases progressively as AR steps grow. This stems from the different draft and target prefix embeddings (Li et al., 2025). Owing to this, the AR models naturally yield divergent predictions, which in turn leads to low $\alpha$. As the AR steps increase, the inputs of the two models gradually converge, thereby improving consistency between their outputs and finally raising the $\alpha$.

To address this, we propose pre-filling a portion (e.g., 5%) of tokens from the target model to ensure a consistent prefix. This does not increase inference latency, as speculative decoding at a low acceptance rate is functionally equivalent to the target model step-by-step decoding (Leviathan et al., 2023). Furthermore, pre-filling improves the overall acceptance rate.

Finally, $p(Y_p)/q(Y_q)$ can be computed with the help of denoising trajectory alignment and token pre-filling to obtain a considerable acceptance rate.

### 3.3 RESAMPLE FROM THE MODIFIED DISTRIBUTION

The simplified illustration of Equation 5 is shown in Figure 5. Unlike discrete form, where $\sum max(0, p(x) - q(x))$ can be directly computed, the analytic expression of $Z$ can't be computed because it involves the integral of the product of a series of Gaussian distributions given by Equation 4 and 3. Therefore, $p'(Y)$ cannot be directly sampled.

To tackle this problem, a viable approach is acceptance–rejection sampling (Casella et al., 2004), which first samples from a proposal distribution, and then calculates a pre-defined rejection threshold $\alpha_s$ and samples $\epsilon \sim U(0,1)$. If $\epsilon < \alpha_s$, the sample is accepted, otherwise it is rejected and sampling from the proposal distribution will be repeated until the sample is accepted.

To realize this sampling method, we first define $\alpha_s$ as:

$$\alpha_s = \frac{p'(Y)}{M \cdot p(Y)}, \tag{10}$$

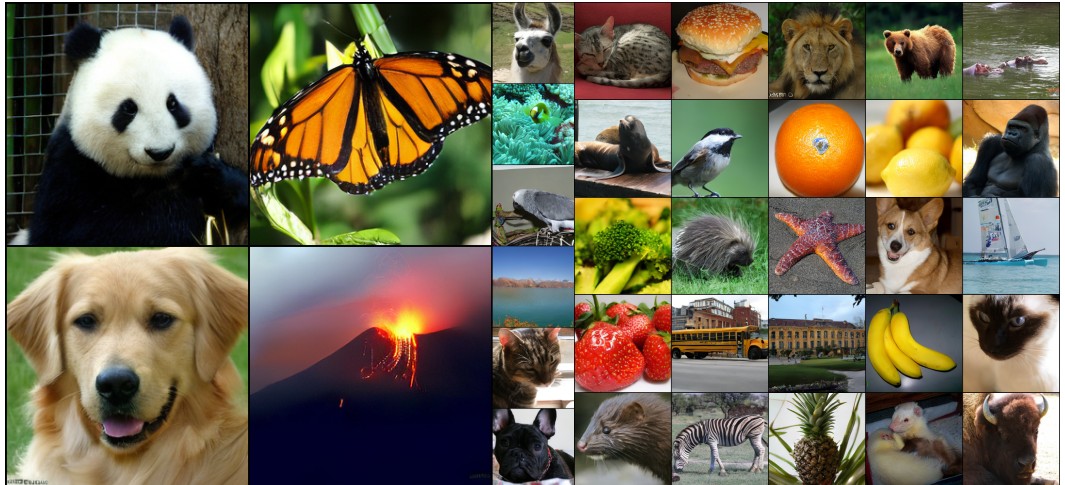

Figure 6: Qualitative results of images generated by MAR with continuous speculative decoding.

where $p(Y)$ is the target distribution, and $M$ is the upper bound factor that holds $M \cdot p(Y) \geq p'(Y)$ for any $Y$. Given $max(0, p(Y) - q(Y)) \leq p(Y)$, $M$ is set to $1/Z$ considering that:

$$p'(Y) = \frac{max(0, p(Y) - q(Y))}{Z} \leq \frac{p(Y)}{Z} \mapsto M \cdot p(Y). \tag{11}$$

We substitute $M = 1/Z$ into Equation 10 to eliminate $Z$:

$$\alpha_s = \frac{max(0, p(Y) - q(Y))/Z}{p(Y)/Z} = \frac{max(0, p(Y) - q(Y))}{p(Y)}. \tag{12}$$

Equation 12 gives the analytic expression of $\alpha_s$ without calculating $Z$. However, in its naive implementation, repetitive diffusion model inference is needed to sample $p(Y)$. It brings heavy extra overhead. To tackle this problem, denoising trajectory alignment is introduced to simplify Equation 12. Accordingly, we have derived the following corollary.

**Corollary 1 (Easy-to-Compute Rejection Threshold)** *With denoising trajectory alignment introduced from Equation 9, the rejection threshold $\alpha_s$ has the easy-to-compute form:*

$$\alpha_s = \frac{max(0, \Sigma \cdot p_\theta(x_0|x_1^p) - q_\theta(x_0|x_1^q))}{\Sigma \cdot p_\theta(x_0|x_1^p)}. \tag{13}$$

See Appendix A for detailed proofs. In this form, $x_0$ is sampled from $p_\theta(x_0|x_1^p)$, which is merely a Gaussian distribution defined by Equation 3, thus avoiding extra model inference. We can compute $\alpha_s$ by gathering $\Sigma$ and sampling $x_0$ from the Gaussian distribution, thereby completing the acceptance-rejection sampling to obtain samples equivalent to those derived from sampling $p'(Y)$.

## 4 EXPERIMENT

### 4.1 IMPLEMENTATION DETAILS

We systematically conduct experiments with open-source continuous visual AR model MAR (Li et al., 2025), xAR (Sucheng et al., 2025) (trained on ImageNet (Deng et al., 2009)) and Harmon (Wu et al., 2025) (trained on more data sources). MAR and xAR are evaluated under $256 \times 256$ resolution. Harmon is evaluated under both $256 \times 256$ and $512 \times 512$ resolutions. MAR-B (208M), xAR-B (172M) and Harmon-0.5B are selected as draft models. MAR-H (943M), xAR-H (1.1B) and Harmon-1.5B are selected as target models. We employ Fréchet Inception Distance (FID) (Heusel et al., 2017), Inception Score (IS) (Salimans et al., 2016), and wall-time speedup on a single NVIDIA A100 GPU as evaluation metrics. More details of experiment settings, ablation studies and quantitative results can be found in Appendix.

| $M_p$ | $M_q$ | $\gamma$ | $\alpha$ | Speedup ratio | | | |
|---|---|---|---|---|---|---|---|
| | | | | bs=1 | bs=8 | bs=128 | bs=256 |
| MAR-H | MAR-B | 32 | 0.19 | **1.44×** | **1.61×** | **2.17×** | **2.33×** |
| MAR-H | MAR-B | 16 | 0.26 | 1.37× | 1.51× | 2.07× | 2.20× |
| MAR-H | MAR-B | 8 | 0.27 | 1.26× | 1.44× | 1.88× | 1.96× |
| MAR-H | MAR-B | 4 | 0.30 | 1.11× | 1.20× | 1.56× | 1.62× |
| xAR-H | xAR-B | 32 | 0.22 | **1.77×** | **2.10×** | **2.52×** | **2.72×** |
| xAR-H | xAR-B | 16 | 0.26 | 1.58× | 2.06× | 2.31× | 2.61× |
| xAR-H | xAR-B | 8 | 0.29 | 1.61× | 1.86× | 2.07× | 2.18× |
| xAR-H | xAR-B | 4 | 0.36 | 1.30× | 1.62× | 1.92× | 2.11× |

Table 1: Results of speedup ratio and acceptance rate $\alpha$ on MAR and xAR under different draft lengths and batch sizes. The bs refers to batch size.

| Resolution | $M_p$ | $M_q$ | $\gamma$ | $\alpha$ | Speedup ratio | | | |
|---|---|---|---|---|---|---|---|---|
| | | | | | bs=1 | bs=8 | bs=16 | bs=32 |
| 256 | Harmon-H | Harmon-B | 32 | 0.17 | **1.47×** | **1.67×** | **1.88×** | **2.05×** |
| | Harmon-H | Harmon-B | 16 | 0.21 | 1.41× | 1.58× | 1.78× | 1.93× |
| | Harmon-H | Harmon-B | 8 | 0.25 | 1.29× | 1.44× | 1.60× | 1.72× |
| | Harmon-H | Harmon-B | 4 | 0.33 | 1.11× | 1.22× | 1.33× | 1.42× |
| 512 | Harmon-H | Harmon-B | 32 | 0.15 | **1.63×** | **1.94×** | **2.23×** | **2.54×** |
| | Harmon-H | Harmon-B | 16 | 0.23 | 1.55× | 1.83× | 2.09× | 2.35× |
| | Harmon-H | Harmon-B | 8 | 0.25 | 1.41× | 1.65× | 1.85× | 2.05× |
| | Harmon-H | Harmon-B | 4 | 0.38 | 1.20× | 1.37× | 1.50× | 1.63× |

Table 2: Results of speedup ratio and acceptance rate $\alpha$ on Harmon under different resolutions, draft lengths, and batch sizes. The bs refers to batch size. Due to CUDA out-of-memory, this set of experiments opt for smaller batch sizes.

| $M_p$ | $M_q$ | w/o CFG | | w/ CFG | |
|---|---|---|---|---|---|
| | | FID↓ | IS↑ | FID↓ | IS↑ |
| MAR-L | | 2.60 | 221.4 | 1.78 | 296.0 |
| MAR-L | MAR-B | 2.59±0.04 | 218.4±3.4 | 1.81±0.05 | 303.7±4.3 |
| MAR-H | | 2.35 | 227.8 | 1.55 | 303.7 |
| MAR-H | MAR-B | 2.36±0.05 | 228.5±2.2 | 1.60±0.05 | 301.6±2.6 |
| MAR-H | MAR-L | 2.34±0.04 | 228.9±2.8 | 1.57±0.04 | 301.4±2.5 |

Table 3: Evaluation of FID and IS on unconditional and conditional generation, compared with original MAR-L and MAR-H models. Our method achieves acceleration while maintaining performance within a reasonable interval.

## 4.2 MAIN RESULTS

**Speedup results.** Table 1 and 2 show the speedup ratio and the overall acceptance rate under different batch sizes, where draft lengths range from 8 to 32. Overall, as the batch size grows, the efficacy of speculative decoding becomes more evident. More specifically, our algorithm achieves an impressive speedup of up to 2.33× on MAR, 2.72× on xAR and 2.54× on Harmon.

**Quantitative results.** The class-conditioned and unconditioned FID and IS metrics for our continuous speculative decoding evaluated on MAR models are shown in Table 3. We conduct multiple experiments and report the average performance and the standard deviation to ensure the reliability of our conclusions. These results demonstrate that our algorithm significantly preserves the quality of generated images, which will be further discussed in subsequent sections. Thus, our approach offers a robust solution for efficient and reliable model inference.

| $M_p$ | $M_q$ | $\gamma$ | $\alpha$ | | $\mathbb{E}\left[\|x^q - x^p\|^2\right]$ | |
|---|---|---|---|---|---|---|
| | | | w/o align | w/ align | w/o align | w/ align |
| MAR-H | MAR-B | 32 | 0.07 | **0.30** | 2.56 | **1.13** |
| MAR-H | MAR-B | 16 | 0.07 | **0.33** | 2.36 | **0.91** |
| MAR-H | MAR-B | 8 | 0.13 | **0.31** | 2.22 | **0.82** |
| MAR-H | MAR-B | 4 | 0.14 | **0.32** | 2.17 | **0.80** |

Table 4: Ablation study on the acceptance rate $\alpha$ and the average distance of each draft and target token with and without denoising trajectory alignment under different draft length $\gamma$.

| $M_p$ | $M_q$ | $\gamma$ | $\alpha$/Speed | | |
|---|---|---|---|---|---|
| | | | 0% | 5% | 15% |
| MAR-H | MAR-B | 32 | 0.25/1.63× | 0.30/1.63× | **0.33**/1.61× |
| MAR-H | MAR-B | 16 | 0.32/1.53× | 0.33/1.52× | **0.34**/1.51× |
| MAR-H | MAR-B | 8 | 0.33/1.47× | 0.31/1.47× | **0.34**/1.44× |
| MAR-H | MAR-B | 4 | 0.31/1.21× | 0.32/1.21× | **0.34**/1.20× |

Table 5: Ablation study on pre-filling ratio. Underline indicates the highest speedup. **Bold** means the highest $\alpha$.

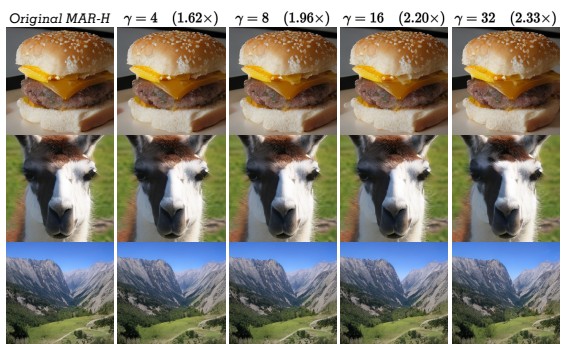

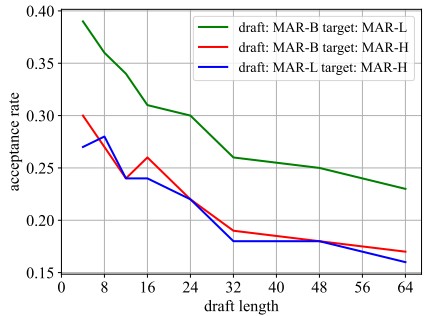

Figure 7: Qualitative Comparison Results. We show the generated images using different draft length $\gamma$.

Figure 8: Plots of acceptance rate $\alpha$ along with draft lengths $\gamma$ on MAR.

**Qualitative results.** Visualization results are shown in Figure 6 and 7 to showcase the image quality generated by our algorithm. Figure 6 demonstrates the results with our algorithm. Figure 7 showcases the results of the original MAR-H with autoregressive step 256 and varying draft lengths $\gamma$. In addition to a significant acceleration, our method can maintain the quality of the generated images, which is consistent with the theoretical proof.

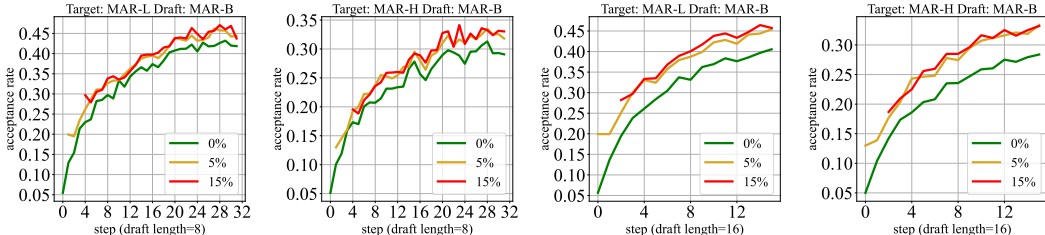

Figure 9: Plots of per-step acceptance rate $\alpha$ under different pre-filling ratios, along with different draft length $\gamma$, averaged on 1000 samples from MAR.

### 4.3 ABLATION STUDY

**The $\alpha$ vs. $\gamma$.** The relationship between acceptance rate $\alpha$ and draft length $\gamma$ on MAR model is depicted in Figure 8. As the length of the draft increases, the acceptance rate tends to decline. This observation suggests that while longer drafts can substantially mitigate inference overhead, they are intrinsically constrained by the capabilities of the draft model itself. Consequently, an increase in the number of draft lengths is associated with greater deviations from the target model's distribution, ultimately leading to reduced acceptance rates.

**Effectiveness of denoising trajectory alignment.** Table 4 shows the acceptance rate and the average distance of each draft and target token with and without our aligned trajectory. The results show that, denoising trajectory alignment reduces the distance between the draft tokens and the target tokens. Before alignment, the distance between them reaches $> 2$, leading to quite small $\alpha$. As the distance is reduced to $\sim 1$, $p(Y)/q(Y)$ increases, further increasing the $\alpha$ to $> 30\%$. Also, denoising trajectory alignment helps to simplify the calculation of $p(Y)/q(Y)$.

**Influence of pre-filled tokens.** The ablation study of pre-filling ratios at $0\%$, $5\%$, and $15\%$ on MAR model is illustrated in Figure 9. Pre-filling can compensate for the low acceptance rates observed during the initial stages of autoregressive sampling and enhance the overall acceptance rate, as shown in Table 5. As the pre-filling ratio increases, the advantages conferred by this approach exhibit diminishing returns.

## 5 CONCLUSION

We present continuous speculative decoding to accelerate continuous visual AR models. We propose denoising trajectory alignment based on proximity in the reparameterization theorem and token pre-filling to enhance the acceptance rate. Acceptance-rejection sampling is introduced with a proper upper bound to sample the modified distribution without analytic expression. The repetitive diffusion model inference is tackled by reusing denoising trajectory alignment. Extensive experiments show that our algorithm achieves over $2\times$ speedup while maintaining the output distribution. We expect our work will provide more thoughts and insights into the inference acceleration with continuous autoregressive models in vision and other domains.

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

APPENDIX

# A  DETAILED PROOFS

We will provide a more detailed process and proof of continuous speculative decoding.

The output of the whole model is composed of the condition from the autoregressive model and the denoising process via the diffusion model. So it can be written as:

$$p(x_{N,0:T}|x_1,\ldots,x_{N-1}) = p(x_{N,T})\prod_{t=1}^{T} p(x_{N,t-1}|x_{N,t},x_1,\ldots,x_{N-1}),$$

where $N$ denotes the number of autoregressive step, $x_1,\ldots,x_{N-1}$ are tokens generated by previous steps, and $t \in [0,T]$ is diffusion timestep. For simplicity, we omit $N$ and $x_{1:N-1}$ and let $Y = x_{0:T}$ to obtain $p(Y) = p(x_T)\prod_{t=1}^{T} p(x_{t-1}|x_t)$.

## A.1  DISCUSSION OF THE APPROXIMATED RATIO

### A.1.1  THE APPROXIMATION OF DIFFUSION DISTRIBUTION

For a token $x$, the acceptance criterion is defined as the ratio of its probability under target distribution to the one under draft distribution, that is, $p(x)/q(x)$. In discrete form, the probability can be directly obtained. But in continuous form, the probability is usually obtained via diffusion process (Ho et al., 2020; Song et al., 2020). Specifically, $x = x_0$ is sampled via reverse diffusion (denoising) process $p(x_T)\prod_{t=1}^{T} p(x_{t-1}|x_t)$. So the probability of the entire denoising process is:

$$p(x_0) = \int_{x_{1:T}} p(x_T)\prod_{t=1}^{T} p(x_{t-1}|x_T)dx_{1:T}, \tag{14}$$

However, Equation 14 is analytically intractable because the product of these complex terms rarely yields a simple, closed-form analytical solution to the high-dimensional integral.

As an alternative to modeling the marginal distribution of a token $x_0$, we model the joint probability of a single fixed path $p(Y)$, where $Y = [x_0, x_1, \ldots, x_T]$, to characterize the probability of each denoising trajectory. The objective of $p(x)$ is to maintain the marginal distribution, while $p(Y)$ aims to maintain the probability of the denoising trajectory itself. Both methods are effective, yet they address distribution maintenance from different perspectives.

However, using $p(Y)$ is impractical for speculative decoding because the acceptance rate is low. To validate this point, we empirically recorded the average value of different kinds of likelihood ratios over 10,000 samples with a draft length of 4, including: (i) the single-path ratio $p(Y)/q(Y)$, (ii) the two-path ratio $p(Y_p)/q(Y_q)$ without denoising trajectory alignment, and (iii) the two-path ratio $p(Y_p)/q(Y_q)$ with denoising trajectory alignment, as shown in Table 6.

| Likelihood ratio | Value | Acceptance rate |
|---|---|---|
| $p(Y)/q(Y)$ | $5.33 \times 10^{-23}$ | 0.0% |
| $p(Y_p)/q(Y_q)$, w/o align | 0.067 | 14% |
| $p(Y_p)/q(Y_q)$, w/ align | 1.86 | 32% |

Table 6: Likelihood ratio comparison of different calculation approaches.

As shown in the first row, the path-space likelihood ratio is extremely small, leading to a 0% acceptance rate. This is because the draft model's trajectory $Y_p$ inherently diverges from the target model's expected trajectory. In each denoising step, samples drawn from the draft model's distribution $q$ are unlikely to fall near $\mu$ of the target distribution $p$, which results in a low single-step ratio $p/q$. As the multi-step denoising process proceeds, the overall $p(Y)/q(Y)$ becomes extremely small.

The second row compares ratios derived from independent trajectories ($Y_p$ and $Y_q$), while the final $x_0$ is generated by the draft model. This $x_0$, without alignment, is highly unlikely to fall near the target

model's target distribution. However, the probability values for the other steps in these trajectories are derived from the model's own path, and thus maintain a relatively reasonable value.

The third row incorporates denoising trajectory alignment. This improvement is twofold: first, our manuscript demonstrates that the expected distance decreases; and second, our analysis shows that the correlation of $p$ and $q$ becomes 1 (common issue 2). Consequently, the samples generated by $q$ have a high probability under $p$, resulting in an increased $p/q$.

For the above reasons, our work adopts a practical approximation for $p(x_0|x_T)/q(x_0|x_T)$: the ratio of the joint probabilities $p(Y_p)/q(Y_q)$, where both $Y_p$ and $Y_q$ share the same $x_0$. This ratio serves as a surrogate for the intractable marginal ratio. By utilizing denoising trajectory alignment, we ensure that $Y_p$ and $Y_q$ are tightly coupled (as discussed in common issue 2), making the likelihood ratio a valid surrogate and ensuring its numerical stability in practical applications.

### A.1.2 How accurately $p(Y_p)/q(Y_q)$ approximates $p(Y)/q(Y)$

We define the joint probability of a single denoising trajectory as $p(Y)$, where $Y$ is the sequence $[x_0, x_1, ...x_T]$. The path space ratio $R = p(Y)/q(Y)$, where $Y = Y_q$, should be an unbiased estimator of $p(Y)$ to ensure that the sampling method preserves the target distribution, that is:

$$\mathbb{E}_{Y_q \sim q}[R] = \int q(Y_q) \frac{p(Y_q)}{q(Y_q)} dY_q = \int p(Y_q) dY_q = 1.$$

However, it is very inefficient. As shown in common issue 1, $p(Y)/q(Y)$ is extremely small, because the trajectory sampled from the draft model may fall into the region where the target model assigns negligible probability mass.

To improve the efficiency, we propose a new estimator, $\tilde{R} := p(Y_p)/q(Y_q)$. $\tilde{R}$ is a biased estimator; its expectation under $q$ is not equal to 1. We analyze the difference between the expectation of $\tilde{R}$ and 1, i.e., its **bias**:

$$B = \mathbb{E}_q[\tilde{R}] - 1 = \mathbb{E}_q\left[\frac{p(Y_p)}{q(Y_q)}\right] - \mathbb{E}_q\left[\frac{p(Y_q)}{q(Y_q)}\right] = \mathbb{E}_q\left[\frac{p(Y_p) - p(Y_q)}{q(Y_q)}\right].$$

We perform a first-order Taylor expansion of $p(Y_p)$ at $Y_q$. Equivalently:

$$p(Y_p) = p(Y_q + (Y_p - Y_q)) \approx p(Y_q) + \nabla p(Y_q)^T (Y_p - Y_q).$$

Substitute into $B$:

$$B \approx \mathbb{E}_q\left[\frac{p(Y_q) + \nabla p(Y_q)^T (Y_p - Y_q) - p(Y_q)}{q(Y_q)}\right] = \mathbb{E}_q\left[\frac{\nabla p(Y_q)^T (Y_p - Y_q)}{q(Y_q)}\right].$$

The magnitude of $B$ is:

$$|B| \approx \left|\mathbb{E}_q\left[\frac{\nabla p(Y_q)^T (Y_p - Y_q)}{q(Y_q)}\right]\right| \leq \mathbb{E}_q\left[\frac{||\nabla p(Y_q)^T||}{|q(Y_q)|}||Y_p - Y_q||\right]. \tag{15}$$

Therefore, the bound of the bias $B$ is proportional to $\mathbb{E}[||Y_p - Y_q||]$, yielding an explicit mean-square error (MSE) bound.

The reduction of expected distance is discussed in Appendix A.2. The expected distance without denoising trajectory alignment is:

$$\mathbb{E}\left[||x_{t-1}^q - x_{t-1}^p||^2\right] = ||\mu_t^q - \mu_t^p||^2 + \text{tr}[\Sigma_t^p + \Sigma_t^q]. \tag{16}$$

The expected distance with denoising trajectory alignment is:

$$\mathbb{E}_{align}\left[||x_{t-1}^q - x_{t-1}^p||^2\right] = ||\mu_t^q - \mu_t^p||^2 + \text{tr}[\Sigma_t^p + \Sigma_t^q - 2\sqrt{\Sigma_t^p \Sigma_t^q}]. \tag{17}$$

The two distances satisfies:

$$\mathbb{E}\left[||x_{t-1}^q - x_{t-1}^p||^2\right] \geq \mathbb{E}_{align}\left[||x_{t-1}^q - x_{t-1}^p||^2\right] \tag{18}$$

The distance over the entire trajectory satisfies:

$$\mathbb{E}\left[||Y_p - Y_q||\right] \leq \sqrt{\sum_{t=1}^T \mathbb{E}||x_t^q - x_t^p||^2}. \tag{19}$$

Substituting into $B$ yields the final first-order error bound:

$$|B| \leq \mathbb{E}_q\left[\frac{||\nabla p(Y_q)^T||}{q(Y_q)}||Y_p - Y_q||\right] \leq \mathbb{E}_q\left[\frac{||\nabla p(Y_q)^T||}{q(Y_q)}\sqrt{\sum_{t=1}^T ||\mu_t^q - \mu_t^p||^2 + \text{tr}[\Sigma_t^p + \Sigma_t^q]}\right] \tag{20}$$

$$|B_{align}| \leq \mathbb{E}_q\left[\frac{||\nabla p(Y_q)^T||}{q(Y_q)}\sqrt{\sum_{t=1}^T ||\mu_t^q - \mu_t^p||^2 + \text{tr}[\Sigma_t^p + \Sigma_t^q - 2\sqrt{\Sigma_t^p\Sigma_t^q}]}\right] \tag{21}$$

The error bound shows that the squared drift difference $||\mu_t^q - \mu_t^p||^2$ and the covariance term $\text{tr}(\Sigma_t^p + \Sigma_t^q)$ or $\text{tr}(\Sigma_t^p + \Sigma_t^q - 2\sqrt{\Sigma_t^p\Sigma_t^q})$ together determine the expected bias incurred by approximating $p(Y)/q(Y)$.

Since the use of denoising trajectory alignment produces the cross term $-2\sqrt{\Sigma_t^p\Sigma_t^q}$, the bias $|B_{align}|$ is typically smaller than $|B|$, thereby supporting a more accurate approximation of $p(Y)/q(Y)$.

### A.1.3 THE EXTENT TO WHICH $p(Y_p)/q(Y_q)$ CAN IMPROVE THE RATIO

Let:

$$\log R = \log p(Y_q) - \log q(Y_q), \tag{22}$$
$$\log \tilde{R} = \log p(Y_p) - \log q(Y_q). \tag{23}$$

The expected difference between the two quantities can be expressed as:

$$\Delta l := \mathbb{E}[\log \tilde{R}] - \mathbb{E}[\log R] = \mathbb{E}[\log p(Y_p)] - \mathbb{E}[\log p(Y_q)]. $$

For each term, we have:

$$\mathbb{E}[\log p(Y_p)] = \mathbb{E}_{Y\sim p}[\log p(Y)] = -H(p), \tag{24}$$
$$\mathbb{E}[\log p(Y_q)] = -H(q) - D_{KL}(q||p). \tag{25}$$

Substituting into $\Delta l$ yields:

$$\Delta l = [-H(p)] - [-H(q) - D_{KL}(q||p)]$$
$$= D_{KL}(q||p) + [H(q) - H(p)]. \tag{26}$$

Empirically, the larger target model is expected to be more capable and to produce more confident (lower-entropy) predictive distributions than the smaller draft model. Therefore, we assume the draft model's entropy $H(q)$ is larger than the target model's entropy $H(p)$. We have:

$$\Delta l = D_{KL}(q||p) + [H(q) - H(p)] \geq 0. \tag{27}$$

It indicates that in the log domain, $\tilde{R}$ exceeds $R$ by $\Delta l$. This implies that our method increases the expected log-ratio, $\mathbb{E}[\log \tilde{R}] - \mathbb{E}[\log R] = \Delta l$, which subsequently leads to a higher expectation of the ratio, $\mathbb{E}[\tilde{R}]$. This explains the empirically observed higher likelihood ratios and increased acceptance rates.

## A.2 DENOISING TRAJECTORY ALIGNMENT

We obtain $x = x_0$ through the denoising process:

$$p(x_{0:T}) = p(x_T) \prod_{t=1}^{T} p(x_{t-1}|x_t), \tag{28}$$

with the conditioned probability distributions as Gaussian approximated by a neural network $\theta$:

$$p_\theta(x_{t-1}|x_t) = \mathcal{N}(x_{t-1}; \mu_\theta(x_t, t), \Sigma_\theta(x_t, t)). \tag{29}$$

Therefore, $p_\theta(x_{t-1}|x_t)$ can be computed using the PDF of the Gaussian distribution. The computation and corresponding notation of $q_\theta(x_{t-1}|x_t)$ are the same.

Empirically, $x_{t-1}$ is obtained by sampling the Gaussian distribution on the right-hand side by **reparameterization**. That is, we first sample $\varepsilon_t \sim \mathcal{N}(0, \mathrm{I})$, and then we obtain the result by scale and shift $x_{t-1} = \sqrt{\Sigma_\theta(x_t, t)} \cdot \varepsilon_t + \mu_\theta(x_t, t)$. To this end, we can compute $p(x)$ and $q(x)$ to obtain the ratio $p(x)/q(x)$.

However, as described in Sec. 3, directly computing the $p(x)$ and $q(x)$ is algebraically correct but may lead to a low acceptance rate due to a distinct denoising trajectory. Thus, we employ the same $\epsilon_t$ in $p(x)$ and $q(x)$ to align their trajectory as closely as possible without affecting the denoising procedure and the results.

**Proof of Theorem 1** Denoising trajectory alignment enhances consistency by reducing the expected inter-sample distance throughout the denoising process. Suppose $x_{t-1}^p = \sqrt{\Sigma_t^p} \cdot \varepsilon_t^p + \mu_t^p$ and $x_{t-1}^q = \sqrt{\Sigma_t^q} \cdot \varepsilon_t^q + \mu_t^q$.

**Without alignment** $(\varepsilon_t^p \neq \varepsilon_t^q)$, let:

$$\begin{aligned} X &= x_{t-1}^q - x_{t-1}^p \\ &= (\mu_t^q - \mu_t^p) + \left( \sqrt{\Sigma_t^p} \cdot \varepsilon_t^p - \sqrt{\Sigma_t^q} \cdot \varepsilon_t^q \right) \\ &= \mu + Y, \end{aligned} \tag{30}$$

where $\mu = \mu_t^q - \mu_t^p$ and $Y = \sqrt{\Sigma_t^p} \cdot \varepsilon_t^p - \sqrt{\Sigma_t^q} \cdot \varepsilon_t^q$. The $||X||^2$ is given by:

$$||X||^2 = X^T X = (\mu + Y)^T (\mu + Y) = \mu^T \mu + \mu^T Y + Y^T \mu + Y^T Y. \tag{31}$$

Therefore:

$$\mathbb{E}\left[||X||^2\right] = \mathbb{E}\left[\mu^T \mu\right] + \mathbb{E}\left[\mu^T Y\right] + \mathbb{E}\left[Y^T \mu\right] + \mathbb{E}\left[Y^T Y\right]. \tag{32}$$

First of all, $\mathbb{E}\left[\mu^T \mu\right] = \mu^T \mu$.

Since:

$$\mathbb{E}\left[Y\right] = \mathbb{E}\left[\sqrt{\Sigma_t^p} \cdot \varepsilon_t^p - \sqrt{\Sigma_t^q} \cdot \varepsilon_t^q\right] = 0 \quad (\text{for } \mathbb{E}\left[\varepsilon_t^p\right] = \mathbb{E}\left[\varepsilon_t^q\right] = 0), \tag{33}$$

we have $\mathbb{E}\left[\mu^T Y\right] = \mathbb{E}\left[Y^T \mu\right] = 0$.

For $\mathbb{E}\left[Y^T Y\right]$:

$$\begin{aligned} Y^T Y &= (\sqrt{\Sigma_t^p} \cdot \varepsilon_t^p - \sqrt{\Sigma_t^q} \cdot \varepsilon_t^q)^T (\sqrt{\Sigma_t^p} \cdot \varepsilon_t^p - \sqrt{\Sigma_t^q} \cdot \varepsilon_t^q) \\ &= (\varepsilon_t^p)^T (\sqrt{\Sigma_t^p})^T \sqrt{\Sigma_t^p} \varepsilon_t^p + (\varepsilon_t^q)^T (\sqrt{\Sigma_t^q})^T \sqrt{\Sigma_t^q} \varepsilon_t^q \\ &\quad - (\varepsilon_t^p)^T (\sqrt{\Sigma_t^p})^T \sqrt{\Sigma_t^q} \varepsilon_t^q - (\varepsilon_t^q)^T (\sqrt{\Sigma_t^q})^T \sqrt{\Sigma_t^p} \varepsilon_t^p. \end{aligned} \tag{34}$$

Note that $(\sqrt{\Sigma_t^p})^T \sqrt{\Sigma_t^p} = \Sigma_t^p$ and $(\sqrt{\Sigma_t^q})^T \sqrt{\Sigma_t^q} = \Sigma_t^q$, we have:

$$Y^T Y = (\varepsilon_t^p)^T \Sigma_t^p \varepsilon_t^p + (\varepsilon_t^q)^T \Sigma_t^q \varepsilon_t^q - (\varepsilon_t^p)^T C \varepsilon_t^q - (\varepsilon_t^q)^T C^T \varepsilon_t^p \quad (C = \sqrt{\Sigma_t^p} \sqrt{\Sigma_t^q}). \tag{35}$$

Since $\mathbb{E}\left[\varepsilon^T M \varepsilon\right] = \text{tr}[M]$ where $\varepsilon \sim \mathcal{N}(0, I_n)$ and $M$ is a matrix, and $\varepsilon_t^p$ and $\varepsilon_t^q$ are independent (expectation of the cross term is 0), we have:

$$\mathbb{E}\left[Y^T Y\right] = \text{tr}[\Sigma_t^p] + \text{tr}[\Sigma_t^q] + 0 + 0 = \text{tr}[\Sigma_t^p + \Sigma_t^q]. \tag{36}$$

Put these results to Equation 32 makes:

$$\boxed{\mathbb{E}_{\varepsilon_t^p \neq \varepsilon_t^q}\left[\left\|x_{t-1}^q - x_{t-1}^p\right\|^2\right] = \left\|\mu_t^q - \mu_t^p\right\|^2 + \text{tr}\left[\Sigma_t^q + \Sigma_t^p\right].} \tag{37}$$

**With alignment** ($\varepsilon_t = \varepsilon_t^p = \varepsilon_t^q$), we have:

$$X = (\mu_t^q - \mu_t^p) + \left(\sqrt{\Sigma_t^p} - \sqrt{\Sigma_t^q}\right)\varepsilon_t$$
$$= \mu + Y\varepsilon_t, \tag{38}$$

where $\mu = \mu_t^q - \mu_t^p$ and $Y = \sqrt{\Sigma_t^p} - \sqrt{\Sigma_t^q}$. We have:

$$\|X\|^2 = (\mu + Y\varepsilon_t)^T(\mu + Y\varepsilon_t)$$
$$= \mu^T \mu + \mu^T Y\varepsilon_t + \varepsilon_t^T Y^T \mu + \varepsilon_t^T Y^T Y\varepsilon_t. \tag{39}$$

Since $\mu^T Y\varepsilon_t$ and $\varepsilon_t^T Y^T \mu$ are scalars, we have:

$$\mu^T Y\varepsilon_t + \varepsilon_t^T Y^T \mu = 2\mu^T Y\varepsilon_t. \tag{40}$$

So:

$$\|X\|^2 = \|\mu\|^2 + 2\mu^T Y\varepsilon_t + \varepsilon_t^T Y^T Y\varepsilon_t. \tag{41}$$

Therefore:

$$\mathbb{E}\left[\|X\|^2\right] = \mathbb{E}\left[\|\mu\|^2\right] + 2\mathbb{E}\left[\mu^T Y\varepsilon_t\right] + \mathbb{E}\left[\varepsilon_t^T Y^T Y\varepsilon_t\right]. \tag{42}$$

Note that $\mathbb{E}\left[\|\mu\|^2\right] = \|\mu\|^2$, and $2\mathbb{E}\left[\mu^T Y\varepsilon_t\right] = 2\mu^T Y\mathbb{E}\left[\varepsilon_t\right] = 0$.

Also, $\mathbb{E}\left[\varepsilon_t^T Y^T Y\varepsilon_t\right] = \text{tr}[Y^T Y]$, we have:

$$\text{tr}[Y^T Y] = \text{tr}[(\sqrt{\Sigma_t^p} - \sqrt{\Sigma_t^q})^T(\sqrt{\Sigma_t^p} - \sqrt{\Sigma_t^q})]$$
$$= \text{tr}[\Sigma_t^p] + \text{tr}[\Sigma_t^q] - 2\text{tr}[\sqrt{\Sigma_t^p \Sigma_t^q}]$$
$$= \text{tr}[\Sigma_t^p + \Sigma_t^q] - 2\text{tr}[\sqrt{\Sigma_t^p \Sigma_t^q}]. \tag{43}$$

Put these results in Equation 39 makes:

$$\boxed{\mathbb{E}_{\varepsilon_t^p = \varepsilon_t^q}\left[\left\|x_{t-1}^q - x_{t-1}^p\right\|^2\right] = \left\|\mu_t^q - \mu_t^p\right\|^2 + \text{tr}[\Sigma_t^p + \Sigma_t^q] - 2\text{tr}[\sqrt{\Sigma_t^p \Sigma_t^q}].} \tag{44}$$

Subtracting Equation 44 from Equation 37 yields:

$$\Delta\mathbb{E} = \mathbb{E}_{\varepsilon_t^p \neq \varepsilon_t^q}\left[\left\|x_{t-1}^q - x_{t-1}^p\right\|^2\right] - \mathbb{E}_{\varepsilon_t^p = \varepsilon_t^q}\left[\left\|x_{t-1}^q - x_{t-1}^p\right\|^2\right]$$
$$= \text{tr}\left[\Sigma_t^q + \Sigma_t^p\right] - \text{tr}[\Sigma_t^p + \Sigma_t^q] + 2\text{tr}[\sqrt{\Sigma_t^p \Sigma_t^q}]$$
$$= 2 \cdot \text{tr}\left[\sqrt{\Sigma_t^q \Sigma_t^p}\right] \geq 0. \tag{45}$$

Furthermore, alignment also simplifies the computation of $\frac{p(x)}{q(x)}$. Note that in Gaussian distribution, we have:

$$p(x) = \frac{1}{(\sqrt{2\pi})^n \sqrt{|\Sigma|}} \exp\left\{\frac{1}{2}(x - \mu)^T \Sigma^{-1}(x - \mu)\right\}$$
$$= \frac{1}{(\sqrt{2\pi})^n \sqrt{|\Sigma|}} \exp\left\{\varepsilon_t^T \varepsilon_t\right\}. \tag{46}$$

Since we have the same $\epsilon_t$ of both $p(x)$ and $q(x)$, the exponential term can be eliminated to obtain:

$$\frac{p(x)}{q(x)} = \frac{\frac{1}{(\sqrt{2\pi})^n \sqrt{|\Sigma_p|}} \exp\left\{\frac{1}{2}(x - \mu_p)^T \Sigma_p^{-1}(x - \mu_p)\right\}}{\frac{1}{(\sqrt{2\pi})^n \sqrt{|\Sigma_q|}} \exp\left\{\frac{1}{2}(x - \mu_q)^T \Sigma_q^{-1}(x - \mu_q)\right\}}$$

$$= \frac{\sqrt{|\Sigma_q|}}{\sqrt{|\Sigma_p|}}. \tag{47}$$

Therefore, for all timesteps $t$:

$$\frac{p(x)}{q(x)} = \frac{p(x_T)\prod_{t=2}^{T} p(x_{t-1}|x_t)}{q(x_T)\prod_{t=2}^{T} q(x_{t-1}|x_t)} \cdot \frac{p(x_0|x_1)}{q(x_0|x_1)}$$

$$= \frac{\prod_{t=2}^{T} \sqrt{|\Sigma_{q,t}|}}{\prod_{t=2}^{T} \sqrt{|\Sigma_{p,t}|}} \cdot \frac{p(x_0|x_1)}{q(x_0|x_1)}$$

$$= \Sigma \cdot \frac{p(x_0|x_1)}{q(x_0|x_1)}, \tag{48}$$

where $\Sigma$ is the cumulative product of $\frac{\sqrt{|\Sigma_{q,t}|}}{\sqrt{|\Sigma_{p,t}|}}$ along the denoising intermediate steps. Also, since $x_0 \sim q(x)$ is verified by the target model, $p(x_0|x_1)$ is not obtained by denoising. It is obtained by substituting $x_0$ into $p(x_0|x_1)$. We keep the two terms since they should be computed separately.

### A.3 ACCEPTANCE-REJECTION SAMPLING

After rejection, we should resample a new output from:

$$p'(Y) = \frac{max(0, p(Y) - q(Y))}{\int_{Y'} max(0, p(Y') - q(Y'))dx'}. \tag{49}$$

But $Z$ is hard to obtain. This integral $Z = \int_{x'} max(0, p(x') - q(x'))dx'$ is difficult to compute and may introduce computation errors if we employ an approximation. This integral also does not have an analytical form.

On the other hand, sampling from proposal distribution $p(Y)$ requires the diffusion loss module to forward for another time, since the entire distribution is determined by all the denoising steps. But in practice, extra model inference introduces heavy overhead and extra latency, and may reduce the speed of speculative decoding, which is harmful for the whole algorithm.

**Proof of Corollary 1**   The introduction of acceptance-rejection sampling can eliminate $Z$ by $M = 1/Z$. The denoising trajectory alignment can simplify the expression and avoid repetitive diffusion model inference. The result is given by:

$$\alpha_s = \frac{max(0, p(Y) - q(Y))/Z}{p(Y)/Z}$$

$$= \frac{max(0, p(Y) - q(Y))}{p(Y)}$$

$$= \frac{max(0, p(x_T)p_\theta(x_0|x_1^p)\prod_{t=2}^{T} p_\theta(x_{t-1}^p|x_t^p) - q(x_T)q_\theta(x_0|x_1^q)\prod_{t=2}^{T} q_\theta(x_{t-1}^q|x_t^q))}{p(x_T)p_\theta(x_0|x_1^p)\prod_{t=2}^{T} p_\theta(x_{t-1}^p|x_t^p)}$$

$$= \frac{max(0, \Sigma \cdot p_\theta(x_0|x_1^p) - q_\theta(x_0|x_1^q))}{\Sigma \cdot p_\theta(x_0|x_1^p)} \tag{50}$$

Afterward, we can obtain the computable results by eliminating the intermediate denoising term denoted as $\Sigma$. The final expression can be derived easily. The modified distribution can be sampled using this approach.

## B  ALGORITHM

Algorithm 1 shows this procedure of continuous speculative decoding algorithm with our implementation of denoising trajectory alignment and acceptance-rejection sampling.

---

**Algorithm 1** ContinuousSpeculativeDecodingStep

---

**Inputs:** $M_p, M_q, prefix$.
▷ Sample $\gamma$ guesses $x_{1,\ldots,\gamma}$ from $M_q$ autoregressively.
**for** $i = 1$ **to** $\gamma$ **do**
    $q_i(Y_q) \leftarrow M_q(prefix + [x_1, \ldots, x_{i-1}])$
    $x_i \sim q_i(Y_q)$
**end for**
▷ Run $M_p$ in parallel, keep the $\epsilon_t$ the same in $M_q$
$p_1(Y_p), \ldots, p_{\gamma+1}(Y_p) \leftarrow$
        $M_p(prefix), \ldots, M_p(prefix + [x_1, \ldots, x_\gamma])$
$\Sigma \leftarrow \frac{\prod_{t=2}^{T} \sqrt{|\Sigma_{q,t}|}}{\prod_{t=2}^{T} \sqrt{|\Sigma_{p,t}|}}$
▷ Determine the number of accepted guesses $n$.
$r_1 \sim U(0,1), \ldots, r_\gamma \sim U(0,1)$
$\frac{p_i(Y_p)}{q_i(Y_q)} \leftarrow \Sigma \cdot \frac{p_i(x|x_1^p)}{q_i(x|x_1^q)}$
$n \leftarrow \min(\{i - 1 \mid 1 \le i \le \gamma, r_i > \frac{p_i(x)}{q_i(x)}\} \cup \{\gamma\})$
▷ Sample the modified distribution via
▷  acceptance-rejection sampling.
**if** $n < \gamma$ **then**
    **repeat**
        $x_t \leftarrow p_n(x|x_1^p)$
        $\alpha_s \leftarrow \frac{max(0, \Sigma \cdot p_n(x_t|x_1^p) - q_n(x_t|x_1^q))}{\Sigma \cdot p_n(x_t|x_1^p)}$
        $\epsilon \sim U(0,1)$
    **until** $\epsilon \le \alpha_s$
**end if**
▷ Return one token from $M_p$, and $n$ tokens from $M_q$.
**return** $prefix + [x_1, \ldots, x_n, x_t]$

---

## C  LIMITATIONS

### C.1  WALL-TIME IMPROVEMENT

As described in Leviathan et al. (2023), the expected walltime improvement is assumed to be:

$$\frac{1 - \alpha^{\gamma+1}}{(1 - \alpha)(\gamma c + 1)}, \tag{51}$$

where $\alpha$ is the acceptance rate of draft tokens, $\gamma$ is the draft length, and $c$ is the inference time ratio between the draft and target models. However, the existing draft model and the target model do not differ significantly in scale. For example, the inference time ratio $c$ of MAR-B (208M) over MAR-H (943M) is 0.38 (bs=128), which is **far more larger** than the number 0.05 or close to 0 mentioned in Leviathan et al. (2023). Increasing the batch size would reduce $c$, which is why our method shows better results on large batch size.

We anticipate that our algorithm will achieve more significant runtime improvements with larger target models, like 7B, 13B, as well as smaller draft models, like 97M, 125M. This direction warrants further investigation in future research.

## D  IMPLEMENTATION DETAILS

We have conducted extensive experiments with open-sourced continuous visual autoregressive model MAR (Li et al., 2025) and xAR (Sucheng et al., 2025) on ImageNet (Deng et al., 2009) $256 \times 256$ generation, and unified model Harmon (Wu et al., 2025) on text-to-image generation. The draft model is chosen from MAR-B (208M), xAR-B (172M) and Harmon-0.5B. The target model is chosen from MAR-H (943M), xAR-H (1.1B) and Harmon-1.5B, respectively. We use official pretrained checkpoints for all models. Since original xAR model is set to predict next cell of the image, the whole image would be generated in 4 steps, we let xAR to autoregressively predict next token at each position, as described as $k = 1$ setting (Sucheng et al., 2025). So the autoregressive step of all the involved draft model is set to 1. However, default MAR models have shown significant results for bidirectional attention in MAR. When target models verify the draft tokens, each output token can be regarded as the last since they can see every previous token. For MAR and xAR, their draft and target models utilize their respective class tokens `[cls]`, which are not shared during the speculative decoding process. Their diffusion loss is not shared either. The batch size ranges in $\{1, 8, 128, 256\}$. The FID and IS are computed on 50k generated images, averaged on ten runs of evaluations. For Harmon models, batch size ranges in $\{1, 8, 16, 32\}$, since larger batch size leads to cuda-out-of-memory. The generation resolution includes both 256 and 512. The draft and target model use their own text embedding respectively. The generation speed is measured on a single NVIDIA A100 GPU.

## E  BORDER IMPACTS

### E.1  BORDER FORMS OF OUTPUT DISTRIBUTION

Various forms of continuous output spaces exist in the visual AR model. For instance, GIVT (Tschannen et al., 2025) and DiCoDe (Li et al., 2024a) employ Gaussian Mixture Models (GMMs) as output distribution. The PDF of token $x$ in GMMs is expressed as:

$$p(x|\theta) = \sum_{k=1}^{K} \pi_k \mathcal{N}(x|\mu_k, \Sigma_k), \tag{52}$$

where $\theta$ represents the model parameters, $\pi_k$ denotes the weights of each Gaussian distribution indexed by $k$, and $\mu_k$ and $\Sigma_k$ indicate the mean and covariance of distribution $k$, respectively. Our method is compatible with GMMs by computing $p(x)$, $q(x)$, and $p'(x)$. The modified distribution $p'(x)$ can still be computed through acceptance-rejection sampling. In practice, considering that current GMM methods cannot achieve competitive performance to MAR and the lack of open-source weights with different model sizes, we haven't conducted related experiments.

This applicability highlights a critical insight: our method relies on an explicit expression for the output distribution to compute $p(x)$, $q(x)$, and subsequently $p'(x)$. As long as the specific functional form of the distribution can be obtained, our algorithm remains universally applicable, regardless of its particular form.

### E.2  MORE VARIANTS OF DIFFUSION MODEL

Other variants of diffusion samplers are still applicable to our method. In this context, we utilize DDIM (Song et al., 2020) as an example since it can be derived from DDPM without additional training.

Theoretically, DDIM (Song et al., 2020) models a conditioned Gaussian distribution during the reverse process, expressed as $p_\theta^{(t)}(x_{t-1}|x_t) = q_\sigma(x_{t-1}|x_t, x_0)$. Its forward (diffusion) process is characterized as a non-Markov process. The term $q_\sigma$ relies on both $x_{t-1}$ and $x_0$, where $x_0$ is predicted by the model $f_\theta$. However, this dependency does not alter the form of the output PDF. The PDF of DDIM given in (Song et al., 2020) remains consistent with that of DDPM and is computed using the expression $p_\theta(x_T) \prod_{t=1}^{T} p_\theta^{(t)}(x_{t-1}|x_t)$. Notably, our algorithm requires an explicit output PDF and does not rely on other properties. In general, the denoising process proceeds through sequential sampling from the previous sample. Let the PDF of initial noise be $p(x_T)$, and let the

PDF for each sampling step be $p(x_{t-1}|x_t, c_{other})$, where $c_{other}$ represents any auxiliary conditions. The final output PDF is then determined as the cumulative product of all intermediate PDFs, namely:

$$p(x_{0:T}|c_{other}) = p(x_T)\prod_{t=1}^{T} p(x_{t-1}|x_t, c_{other}). \tag{53}$$

This expression is valid across all diffusion models, regardless of specific samplers or diffusion processes like DDIM, Rectified Flow, etc.

Practically, the original implementation of MAR is realized through DDPM. Converting DDPM to DDIM does not require additional training procedure (Song et al., 2020). We adopt the DDPM to the DDIM in the diffusion loss of MAR and present the relevant results in Table 7. We use MAR-B as the draft model and MAR-H as the target model. The batch size is 256, and the denoising step of DDIM is set to 100. Our method and formulas remain applicable to other variants of diffusion models and continue to demonstrate good performance.

| draft length $\gamma$ | 4 | 8 | 16 | 32 |
|---|---|---|---|---|
| speed up ratio | 1.58 | 1.92 | 2.15 | 2.30 |
| acceptance rate $\alpha$ | 0.26 | 0.24 | 0.21 | 0.18 |

Table 7: Speed up ratio and acceptance rate on MAR with DDIM as diffusion loss under different number of draft lengths.

### E.3 BORDER IMPLEMENTATION DOMAINS

Continuous speculative decoding is not confined to autoregressive image generation tasks. It can also be applied to other domains, such as autoregressive audio or video generation in continuous spaces and broader tasks and scenarios within the continuous domain. Due to the limitations of currently available models, we have not been able to identify usable models beyond the domain of autoregressive image generation. However, as an algorithm without additional training or performance loss, speculative decoding remains one of the most optimal acceleration methods. We hope that continuous speculative decoding can provide more valuable insights and ideas for researchers in various fields and stimulate further research on speculative decoding across diverse domains.

## F INFORMATION ABOUT USE OF AI ASSISTANTS

In the preparation of this work, we employ AI assistants to assist with refining academic language. The AI tools were used solely for improving clarity, grammatical correctness, and syntactic efficiency—tasks analogous to those performed by a human editor or linter. All conceptual contributions, technical claims, and critical analysis remain the authors' own.

## G ADDITIONAL EXPERIMENTS

### G.1 EFFECTIVENESS OF DRAFT & VERIFICATION.

Figure 10a demonstrates the comparative generation results using a pure draft model versus the draft & verification paradigm. During the draft & verification process, those suboptimal token regions in the draft results are systematically identified and substituted with higher-quality tokens during verification by the target model. This approach maintains the overall compositional integrity while significantly enriching the detail and quality of the generated images.

### G.2 ACCEPTANCE-REJECTION SAMPLING

Standard acceptance-rejection sampling favors large total variance $p(x) - q(x)$ for high acceptance rate, while speculative decoding prefers small differences for better draft model alignment. Therefore, we observe the practical total inference wall-time of target MAR model during the whole

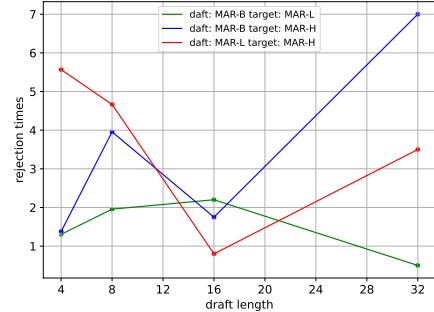

(a) Comparison on pure draft (left) and verified (right) generation results. Regions of rejected tokens are roughly marked out.

(b) Empirical rejection times in acceptance-rejection sampling algorithm of the rejection phase.

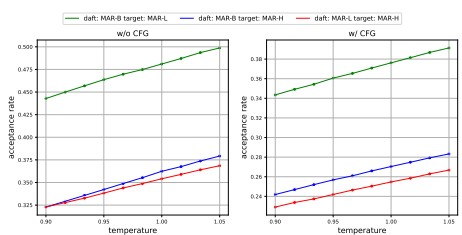

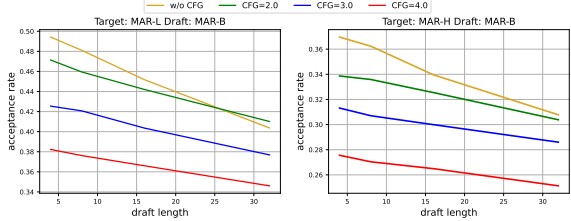

(c) Temperature influence on the acceptance rate. Left: without CFG. Right: with CFG.

(d) CFG scale has has a significant impact on the acceptance rate under different number of draft lengths.

Figure 10: Ablation studies on various experiment factors conducted on MAR models.

| Draft length $\gamma$ | 4 | 8 | 16 | 32 |
|---|---|---|---|---|
| Target model runtime | 64s | 56s | 53s | 51s |
| Rejection-sampling rutime | 0.0378s | 0.0189s | 0.0095s | 0.0047s |

Table 8: Overall runtime of target model and acceptance-rejection sampling during the whole speculative decoding process at different draft length $\gamma$. Batch size is set to 1.

speculative decoding process at batch size=1, and the wall-time and sampling steps of acceptance-rejection sampling. As show in Table 8, the rejection-sampling actually only accounts for a quite small fraction of the model runtime. Overall the inference speed is improved by speculative decoding. Figure 10b illustrates the relationship between the rejection times and the draft length. Empirically, acceptance-rejection sampling often requires only a few sampling steps. The runtime consumed by this sampling process is negligible compared to the overall model inference time.

### G.3 TEMPERATURE

Temperature $\tau$ is a crucial hyperparameter during the denoising process in MAR. The temperature setting affects the consistency between the outputs of the draft and target models. Figure 10c illustrates the impact of the temperature $\tau$ on the acceptance rate during the generation process. The number of drafts length is set to 8. The temperature influences the PDF of the final output distribution; a lower temperature may result in a sharper distribution, while a higher temperature may lead to a flatter distribution. The ratio $p(x)/q(x)$ can be influenced based on this.

### G.4 CLASSIFIER-FREE GUIDANCE

Figure 10d illustrates the relationship between draft length and the acceptance rate under different CFG scales. As the CFG scale increases, there is an overall trend of decreasing acceptance rates. This trend remains consistent mainly across each draft length. This phenomenon may indicate that

the inconsistency between the draft and target models may increase as class guidance strengthens, reducing the acceptance rate.

### G.5 COMPARISON WITH MASKED GENERATION

MAR can achieve generation acceleration by generating multiple tokens per step. However, the cost of acceleration comes with a noticeable performance loss. Speculative decoding typically provides a 2× speedup, consistent with the conclusion drawn in the application of LLMs (Leviathan et al., 2023). The performance is well maintained, as theoretically proved. In contrast, while masked generation can achieve a higher acceleration ratio (up to 10× when the number of masks reaches 32), it cannot maintain performance, as shown in Table 9. We show the speedup ratio on MAR-H with 256 batch size. As the acceleration ratio increases, the model's performance suffers significant degradation.

| # mask | 2 | 4 | 8 | 16 | 32 | 64 |
|--------|------|------|------|------|-------|-------|
| speed up | 1.99 | 3.49 | 5.54 | 7.85 | 10.02 | 11.49 |
| FID | 1.56 | 2.37 | 3.66 | 4.99 | 17.43 | 59.07 |

Table 9: Speed-up ratio and FID under different mask generation steps. Larger mask generation step can bring the model a better speedup, but it also leads to significant performance degradation.

### G.6 VISUALIZATION OF ACCEPTANCE

We visualize the acceptance and rejection region of each position through a 2D heatmap. As shown in Figure 11, dark green blocks represent accepted tokens, while light green blocks represent rejected. We observe that tokens representing backgrounds or regions with simpler textures tend to be accepted. In contrast, more detailed positions are more likely to be rejected.

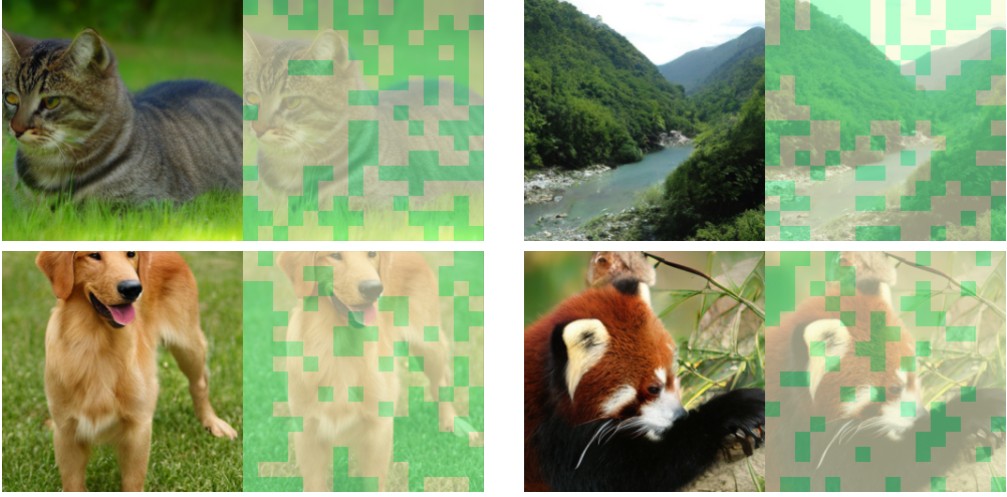

Figure 11: Visualizations of accepted token heatmap. Dark green: accepted. Light green: rejected.

### G.7 FAILURE MODES

We present a visualization of the images associated with the observed failure modes and success modes. The resulting visualization is presented in Figure 12. Subfigures (a) and (b) depict the failure modes, while subfigures (c) and (d) illustrate the success modes. Our analysis reveals that the failure modes are predominantly localized in regions exhibiting high levels of detail and intricate texture representation. Subfigures (a) and (b) are characterized by rich details and fine textures, where the limited capacity of the draft model results in generated content that is below an acceptable quality threshold. In contrast, subfigures (c) and (d) possess comparatively lower complexity in terms of

detail. The two subfigures include substantial background area devoid of pronounced details. The acceptance rate is substantially elevated within these less-detailed regions.

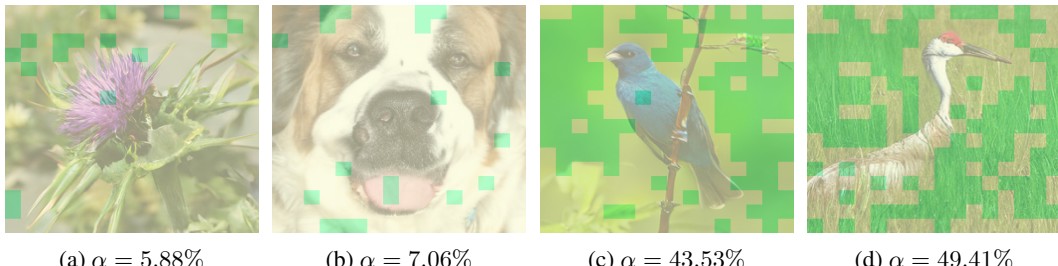

(a) $\alpha = 5.88\%$      (b) $\alpha = 7.06\%$      (c) $\alpha = 43.53\%$      (d) $\alpha = 49.41\%$

Figure 12: Subfigure (a) and (b): failure modes. Subfigure (c) and (d): success modes. The visualizations reveal that the failure modes are predominantly localized in regions exhibiting high levels of detail and intricate texture representation.

## H  MORE QUALITATIVE RESULTS

In Figure 13, 14, 15, 16 and 17, we provide more additional images generated under our continuous speculative decoding by MAR compared with the target model only. While the target model has achieved satisfactory quality in generating realistic and high-fidelity images, our continuous speculative decoding can show comparable performance, similar generation results, and much faster inference speed.

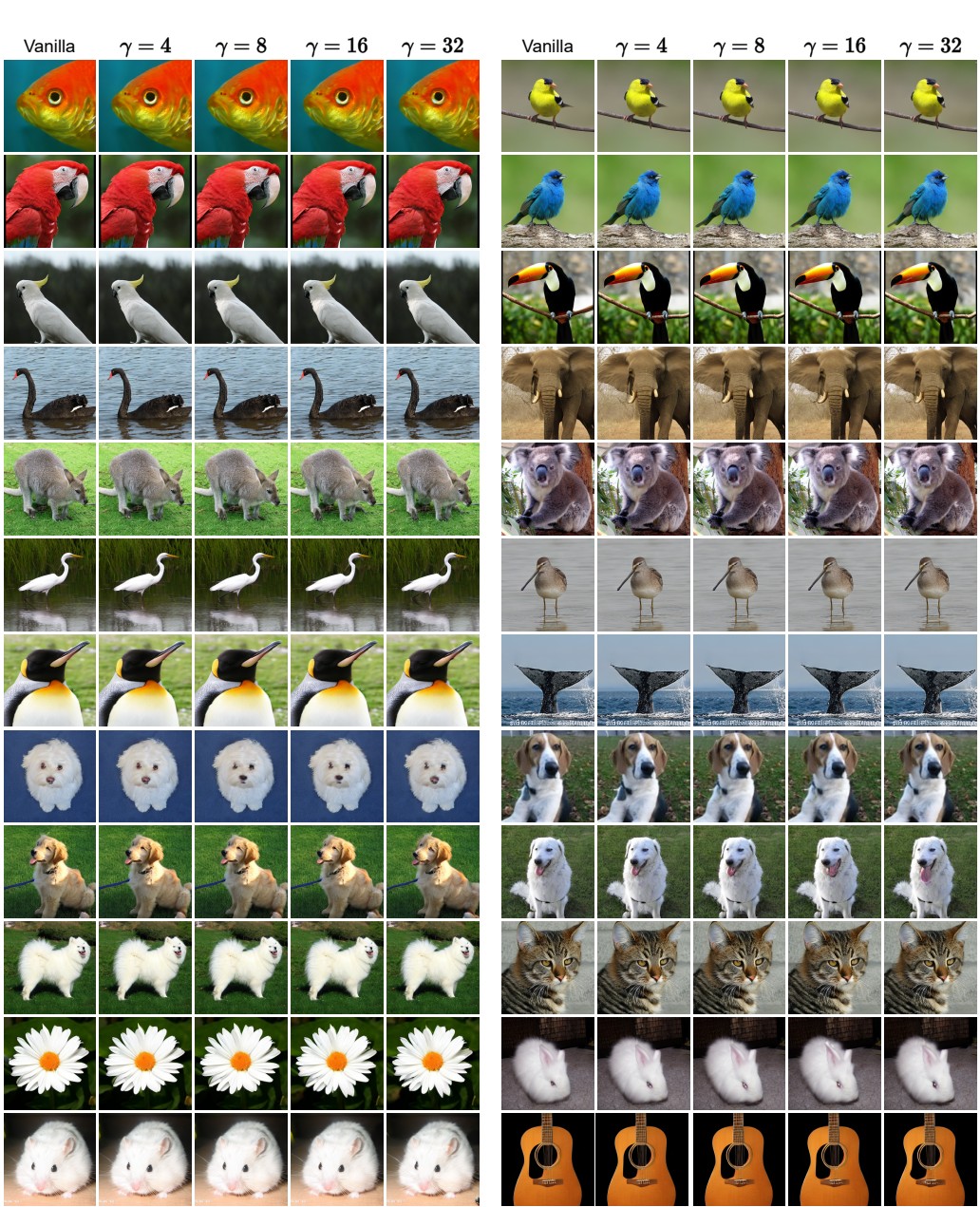

Figure 13: Visual quality with increasing draft length $\gamma$ compared with vanilla target model only generation. *Best viewed zoom-in.*

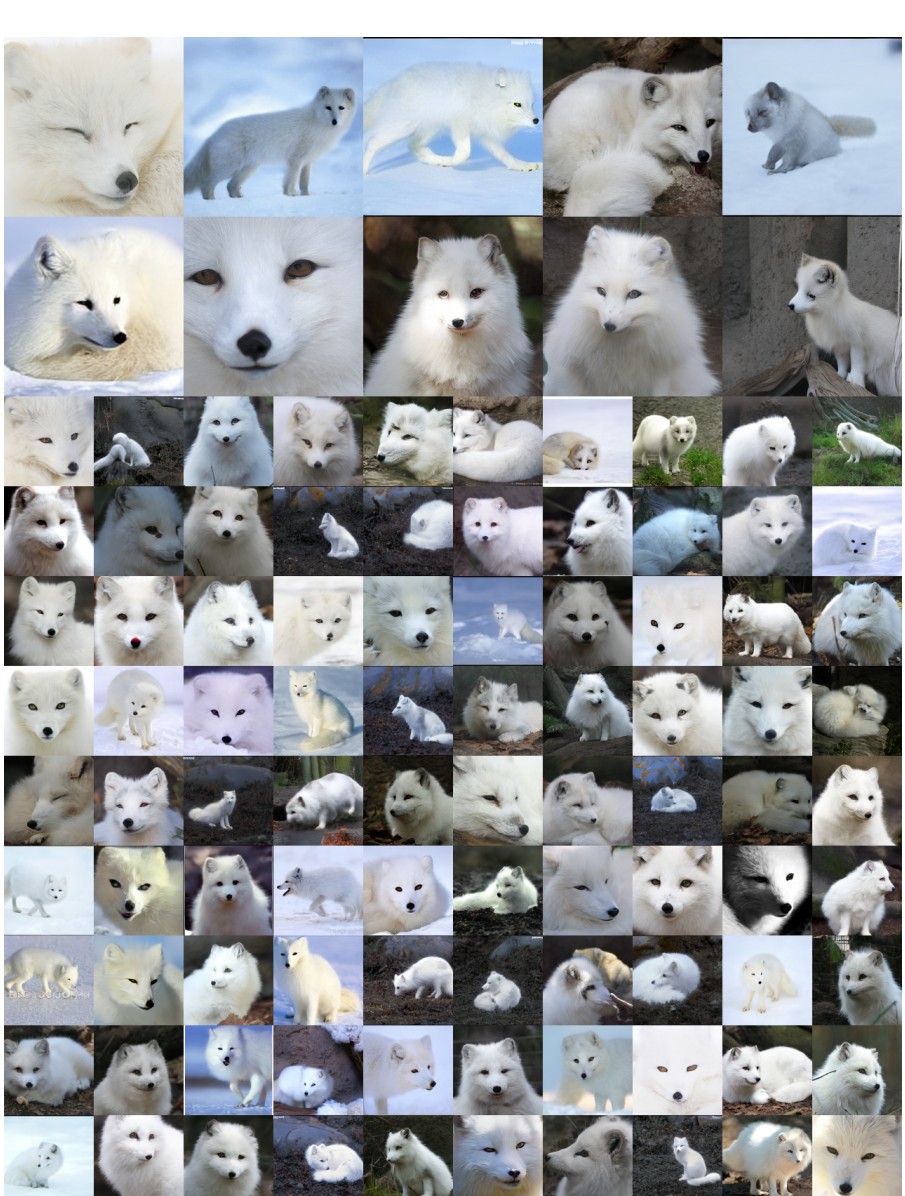

Figure 14: Visualization examples under $\gamma = 4$. Class label: arctic fox (297).

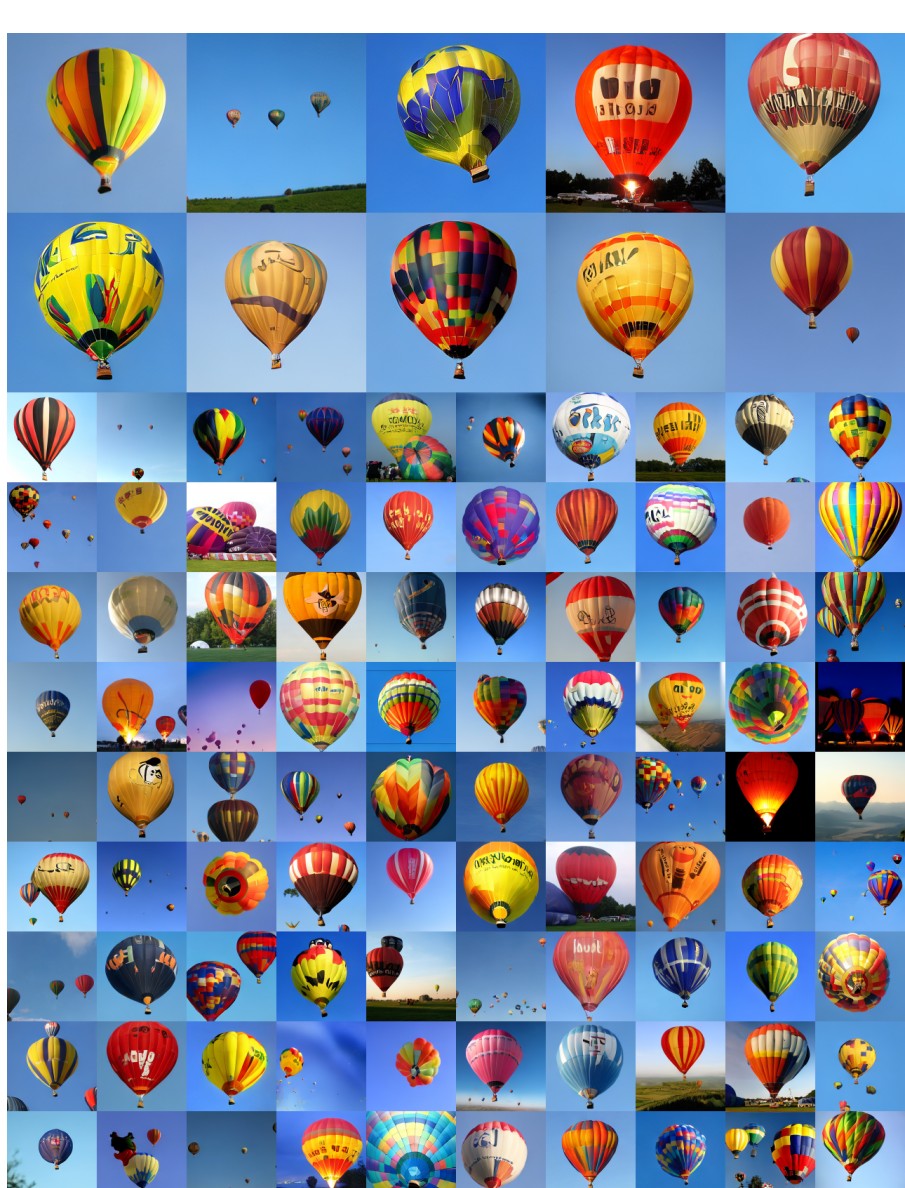

Figure 15: Visualization examples under $\gamma = 8$. Class label: balloon (417).

Figure 16: Visualization examples under $\gamma = 16$. Class label: ice cream (928).

Figure 17: Visualization examples under $\gamma = 32$. Class label: volcano (980).

