# OpenReview forum: "Continuous Speculative Decoding for Autoregressive Image Generation"
_ICLR.cc/2026/Conference — Submitted to ICLR 2026_

### Official Review · Reviewer_gror · 2025-10-30

**Soundness:** 2
**Presentation:** 2
**Contribution:** 3
**Rating:** 2
**Confidence:** 4

**Summary:**

This paper proposes a new speculative decoding algorithm, especially for diffusion-head-based continuous tokenizer AR models such as MAR. Specifically, to realize the speculative sampling process in continuous space, it (i) first simplifies the likelihood ratio evaluation (p/q) using a one-step denoising ratio to determine the acceptance rate, and (ii) performs rejection-sampling-based resampling from the residual distribution. The experimental results show that this method accelerates the decoding process by ~2x while maintaining image quality.

**Strengths:**

- The paper is well written and easy to understand.
- To my knowledge, this is almost the first work that tries to adopt speculative decoding in continuous space.
- The results shows promising acceleration ratios in various settings.

**Weaknesses:**

I think this paper build upon some critical theoretical flaws and is not correct Speculative Decoding (SD) :
- Eq. (2) is not correct. The left side, $p(x_0|x_T)$, is the marginal probability of $x_0$, but the right side is actually the joint probability of the trajectory, $p(x_{0:{T-1}}|x_T)$. The correct relation is $p(x_0\mid x_T)=\int \Pi_{t=1}^T p(x_{t-1}|x_t) dx_{1:T-1}$. This invalidates all following discussions and methods, as they all rely on an incorrect $p/q$ ratio.

- If the authors intended to perform SD on the path space $p_{path}(x_{0:T-1}|x_T)$ and pick $x_0$ from an accepted/resampled path, then it would be correct. However, the current algorithm does not actually do this.
  - Correct path-space SD requires the likelihood ratio $p_{path}(Y)/q_{path}(Y)$ on a **single fixed path** $Y = [x_0, x_1, .. x_T]$, which is generated by the draft model $q_{path}$.
  - However, "noise alignment" actually yields two different paths, $Y_p$ and $Y_q$. They typically do not follow the same path, even if we use the same noise at the start and during denoising. Equation (8) is just the ratio of self-likelihoods under two different self-paths, and is neither (i) the correct $p(x_0)/q(x_0)$ nor (ii) the correct $p_{path}(Y)/q_{path}(Y)$. Thus, the lossless property of SD cannot be achieved, and the resampling process is also incorrect because it depends on the wrong $p/q$ ratio.
- I think this paper is closer to lossy SD, which tries to approximate the intractable $p(x)/q(x)$ by a practically feasible surrogate $p(x_0|x^p_1)/q(x_0|x^q_1)$ (which is the last-step denoising difference under different self-paths), and paper shows that final image quality can still be almost maintained. If so, the motivation for the proposed methods is weak, because they focus mostly on computation tricks and the sampling process for exactly recovering $p(x)$ by SD.

**Questions:**

- While the Target model and Draft model are different AR models, do they share the same diffusion model head?

- In Tables 1 and 2, why does the acceptance rate decrease as the draft length increases?

- In Tables 1 and 2, why does the speed-up increase as the batch size increases? Using larger batch size increase computational overhead and typically increase latency.

- In Figures 7 and 12, why do the 'original' and 'SD' samples show sample-level identity? SD basically just guarantees distribution-level identity  with original AR decoding, not individual sample-level identity.

- In Table 4, why does 'alignment' increase the acceptance rate? The acceptance rates should only depends on the divergence between the target model ($p$) and the draft model ($q$), actually $1 - \text{TotalVariation}(p, q)$ [1]. Does this imply that the alignment process itself modifies the draft distribution $q$?

[1] SpecTr: Fast Speculative Decoding via Optimal Transport; NeurIPS 2023

---

> ### Author Response · Authors · 2025-11-21
> **To reviewer gror**
>
> We appreciate the reviewer's thoughtful feedback. Below, we address each concern in detail.
>
> **Weaknesses 1: Quesiton on Equation (2)**
>
> Please see **Common issue 1** for details.
>
>
>
> **Quesiton 1: Diffusion model head**
>
> They use separate diffusion model heads.
>
>
>
> **Quesiton 2: Acceptance rate decrease as the draft length increases**
>
> As the draft length increases, subsequent draft tokens gradually accumulate larger deviations from the target model’s expected distribution. These accumulated deviations reduce the probability that later draft tokens meet the target model’s verification criterion, ultimately leading to a lower overall acceptance rate.
>
>
>
> **Quesitons 3: Speed-up increase as the batch size increases**
>
> Our reported "speed-up ratio" quantifies the acceleration relative to the baseline target model, distinct from the absolute latency, which refers to the total generation time. As discussed in Appendix C.1, the key insight is system efficiency. The smaller draft model requires less computation per batch than the larger target model. As the batch size increases, the target model’s latency increases much faster than the draft model’s, as shown in the table below (measured on an A100 GPU). This widening latency gap increases the speed-up ratio, even as the absolute latency increases for both models.
>
> | **Batch size**  | **1** | **8** | **64** | **128** | **256** |
> | --------------- | ----- | ----- | ------ | ------- | ------- |
> | MAR-H (target)  | 70s   | 82s   | 146s   | 232s    | 386s    |
> | MAR-B (draft)   | 44s   | 46s   | 65s    | 85s     | 127s    |
> | Latency ratio c | 0.63  | 0.56  | 0.45   | 0.37    | 0.33    |
>
>
>
> **Question 4: Sample-level identity**
>
> We emphasize that samples generated by our method show subtle yet distinct differences, rather than strictly sample-level identity. The differences require careful inspection. Specific differences are highlighted across the referenced figures below:
>
> 1. Figure 1:
>    - Bottom-left panel (Dog): The dog in the bottom-left panel displays distinct back fur patterns in the arrangement of light and brown stripes.
> 2. Figure 12:
>    - Fifth row from the bottom (Right-side dog sample): The decorative accessory around the dog’s neck changes in shape as draft length increases.
>    - Fourth row from the bottom (Right-side dog with tongue out): The tongue’s position and contour vary—for example, the tongue extends slightly farther outward.
>    - Second row from the bottom (Left-side flower sample): Petal edges show subtle irregularities.
>    - Last row (Right-side guitar sample): The black decorative patterns on the guitar’s body differ in spacing.
>
>
>
> **Question 5: The effect of alignment**
>
> Please see **Common issue 2** for details.

---

> > ### Comment · Reviewer_gror · 2025-11-24
> > **Thanks for the response**
> >
> > Thank you for the rebuttal. My concerns about Q2-4 is resolved. Also I appreciate the author's clarification that this is indeed an approximation. However, the current response does not alleviate my concerns regarding W1.
> >
> > (0) First, please clarify exactly what is being approximated.
> > Is the goal to preserve the path distribution $P_{path}(Y)$ as stated in the response to oZF2, or is it to approximate the marginal $p(x_0)$ as suggested in Common Issue 1 and Line 238 of the main text? If the goal is to preserve the path distribution, the low acceptance rate of this likelihood ratio is not a "problem" but rather an inherent characteristic of high-dimensional continuous spaces. Consequently, an artificially improved acceptance rate would simply imply an inaccurate recovery of the path distribution.
> >
> > (1) I will assume the objective is to recover the sample distribution for $p(x_0)$, as implied in the main text.
> > The authors claim that they approximate $p(x_0)/q(x_0)$ using the proposed surrogate because the true ratio is computationally intractable due to integration, and the acceptance rate based on $P_{path}(Y)/Q_{path}(Y)$ is "low." However, achieving a higher acceptance rate is trivial if one is willing to sacrifice the accuracy of the acceptance probability approximation (like all other lossy SD). Therefore, to justify the motivation for this methodology, the authors must demonstrate the **validity of this approximation** through theoretical analysis or, at the very least, empirical experiments. The current version of the paper lacks this entirely.
> >
> > (2) While the authors acknowledge that this is an approximation, Eq. 4 and the subsequent text treat it as if it were $p(x)$ again. This notation is very confusing and should not be used without clear rationalization.
> >
> > Below are the minimum experiments I believe are necessary to justify the paper's claims:
> >
> > * Please add experiments demonstrating how accurately $P_{path}(Y_p)/Q_{path}(Y_q)$ approximates $p(x_0)/q(x_0)$. If the approximation is accurate, a discussion on why this is the case is required. If it is not, you must discuss why the final generation quality is preserved despite the inaccurate approximation.
> > * Despite this being a lossy Speculative Decoding (SD) method, quantitative experiments regarding accuracy are severely lacking, with the focus remaining primarily on speed. Experiments in more diverse settings are required to quantify "how much" accuracy is actually preserved.

---

> > > ### Author Response · Authors · 2025-11-27
> > > **Response to reviewer gror**
> > >
> > > We sincerely thank the reviewer for the continued engagement and valuable feedback. We understand that your primary remaining concern relates to the approximation quality of our method.
> > >
> > > **Q: Discussion about the approximation accuracy**
> > >
> > > We clarify that **the joint probability of the denoising path**, $P_{path}(Y)$, is approximated, not the marginal distribution. **Common Issues 3 and 4** discuss in detail both the accuracy with which $P_{path}(Y_p)/Q_{path}(Y_q)$ approximates $P_{path}(Y)/Q_{path}(Y)$ and the extent to which our approximation improves the likelihood ratio.
> > >
> > > We have revised the notation statements in the manuscript to explicitly claim the consideration of the joint distribution $P_{path}(Y)$ and ensure consistent usage in the subsequent notations.
> > >
> > > **Q: More quantitative experiments regarding accuracy**
> > >
> > > We report additional quantitative results regarding generation quality to assess the accuracy of our method. Specifically, we provide the following evaluations: (1) FID and IS metrics for xAR (with an AR step of 256) on ImageNet; (2) the FID for the text-to-image model Harmon on MSCOCO and MJHQ; (3) the CLIPScore for Harmon on MSCOCO; and (4) metrics for Harmon on Geneval, as shown in the tables below. These results demonstrate that our method preserves the quality of generated images. Even though the likelihood ratio is approximated within the algorithm, as discussed in the previous question, this approximation introduces only minimal bias and largely preserves the quality of image generation.
> > >
> > > (1) FID and IS metrics for xAR on ImageNet:
> > >
> > > | Target model | Draft model | FID    | IS |
> > > | ------------ | ----------- | --------- |-------|
> > > | xAR-L     | -           | 1.87      |274.8|
> > > | xAR-L     | xAR-B    | 1.88     |270.4|
> > > | xAR-H     | -           | 1.79      |288.9|
> > > | xAR-H     | xAR-B    | 1.82      |286.1|
> > > | xAR-H     | xAR-L    | 1.78      |287.7|
> > >
> > >
> > > (2) FID for Harmon on MSCOCO and MJHQ:
> > >
> > > | Target model | Draft model | MSCOCO    | MJHQ |
> > > | ------------ | ----------- | --------- |-------|
> > > | Harmon-H     | -           | 8.39      |5.15|
> > > | Harmon-H     | Harmon-B    | 8.38      |5.13|
> > >
> > >
> > > (3) CLIPScore for Harmon on MSCOCO:
> > >
> > > | Target model | Draft model | CLIPScore |
> > > | ------------ | ----------- | --------- |
> > > | Harmon-H     | -           | 34.8      |
> > > | Harmon-H     | Harmon-B    | 34.7      |
> > >
> > > (4) Harmon Evaluation of Geneval:
> > >
> > > | Target model | Draft model | Single Obj. | Two Obj. | Counting | Colors | Position | Color Attri. | Overall |
> > > | ------------ | ----------- | ----------- | -------- | -------- | ------ | -------- | ------------ | ------- |
> > > | Harmon-H     | -           | 0.99        | 0.86     | 0.66     | 0.85   | 0.74     | 0.48         | 0.76    |
> > > | Harmon-H     | Harmon-B    | 0.99        | 0.83     | 0.66     | 0.86   | 0.74     | 0.44         | 0.75    |

---

### Official Review · Reviewer_EoYy · 2025-11-02

**Soundness:** 3
**Presentation:** 3
**Contribution:** 3
**Rating:** 6
**Confidence:** 3

**Summary:**

The manuscript introduces Continuous Speculative Decoding (CSD) to accelerate inference for continuous visual autoregressive (AR) image generators whose per-token distributions are implemented via diffusion. The method mirrors discrete speculative decoding (draft-and-verify) but tackles two problems for continuous in continuous setting: 1) very low acceptance due to draft-verify distribution mismatch and 2) the intractability of sampling from the modified reject distribution. To this end, the manuscript proposes denoising trajectory alignment (sharing the diffusion noise across draft/target) and token pre-filling (seed the prefix with a small fraction of target tokens) to raise acceptance, and use acceptance–rejection sampling with a derived upper bound to sample from the modified distribution without evaluating a difficult integral. On various models, they report up to ~2.3–2.7× speedups at larger batch sizes while keeping image quality roughly unchanged.

**Strengths:**

- It is the first to apply speculative decoding in continuous settings.
- Discovers practical problems and handles them by proposing techniques to raise acceptance in practice.
- The method is training-free, facilitating practicality in deployment.

**Weaknesses:**

- Most speedups occur at large verification batch sizes, whereas bsz=1 shows diminished speedups. For many interactive or small-batch generation workloads, the practical acceleration may be lower than the headline.
- Quality preservation is assessed primarily with FID/IS. There is no evaluation of text-image faithfulness, such as CLIPScore or GenEval.

**Questions:**

I wish to defer this to the discussion phase.

---

> ### Author Response · Authors · 2025-11-21
> **To reviewer EoYy**
>
> We thank the reviewer for the valuable feedback. We address each concern below.
>
> **Weakness 1: Practical acceleration**
>
> We appreciate the reviewer’s suggestion. In real-world service deployments, batching is universally used to optimize resource efficiency. User requests are aggregated into small-to-medium batches, which is sufficient to realize gains from batch inference. This approach avoids the inefficiency of batch size 1 inference while still meeting interactive latency requirements.
>
>
>
> **Weakness 2: Evaluation of text-image faithfulness**
>
> Thank you for highlighting this point. We conduct experiments on the Harmon (text2img) to evaluate CLIPScore on COCO and GenEval benchmark, as shown in the tables below. These evaluations confirm that our method maintains text-image faithfulness and generation quality while accelerating inference.
>
> Evaluation of CLIPScore:
>
> | Target model | Draft model | CLIPScore |
> | ------------ | ----------- | --------- |
> | Harmon-H     | -           | 34.8      |
> | Harmon-H     | Harmon-B    | 34.7      |
>
> Evaluation of Geneval:
>
> | Target model | Draft model | Single Obj. | Two Obj. | Counting | Colors | Position | Color Attri. | Overall |
> | ------------ | ----------- | ----------- | -------- | -------- | ------ | -------- | ------------ | ------- |
> | Harmon-H     | -           | 0.99        | 0.86     | 0.66     | 0.85   | 0.74     | 0.48         | 0.76    |
> | Harmon-H     | Harmon-B    | 0.99        | 0.83     | 0.66     | 0.86   | 0.74     | 0.44         | 0.75    |

---

> > ### Comment · Reviewer_EoYy · 2025-11-28
> >
> > Thank you for the experimental clarifications; my concerns are resolved.
> >
> > Although I refrain from further increasing the score due to my confidence in the topic, I wish to state that I lean towards Accept.

---

### Official Review · Reviewer_oZF2 · 2025-11-02

**Soundness:** 1
**Presentation:** 3
**Contribution:** 3
**Rating:** 4
**Confidence:** 4

**Summary:**

This paper introduces a new framework extending speculative decoding from discrete to continuous AR models, particularly diffusion-based visual AR systems. Traditional speculative decoding on continuous distributions poses two key obstacles: (1) inconsistent output distributions between draft and target models, which lead to low acceptance rates, and (2) the absence of an analytic form for the modified distribution due to complex normalization integrals. To tackle these, the authors propose denoising trajectory alignment and token pre-filling, and  they derive an acceptance–rejection sampling algorithm with a tractable upper bound, eliminating the need for intractable integrals, and reuse trajectory alignment to compute rejection thresholds efficiently. The method integrates seamlessly into existing continuous AR models without retraining or architectural changes.

**Strengths:**

The paper’s strongest aspect is that it identifies and articulates a genuinely nontrivial gap between discrete speculative decoding and continuous, diffusion-based AR generation, and then proposes a concrete recipe to fill it. Recognizing that the core obstacle is distributional inconsistency is an original reframing, and it is precisely this reframing that motivates the two key ideas, denoising trajectory alignment and token pre-filling, rather than importing the discrete algorithm verbatim.

In terms of originality, this is not just “we do speculative decoding for images now,” but “we make speculative decoding compatible with diffusion-style tokenization,” which, to my knowledge, is not handled in earlier speculative work on discrete visual AR or in speedup methods that simply prune or distill the denoiser. The proposal to align draft/target diffusion paths via shared reparameterization noise is also a creative repurposing of a well-known trick (noise sharing) to solve an acceptance-rate issue rather than a sample-quality issue, which is a nice shift of perspective.

**Weaknesses:**

The acceptance ratio in Eq. (2) is written as if it used the marginal, but what is actually computed is the factorized reverse-diffusion path probability, i.e. a joint over a sampled trajectory. This is only equal to the desired marginal when you integrate out intermediate states, which the method does not do.Because the same approximation is applied to both p and q, the authors hope the error cancels, but that cancellation is only heuristic and depends critically on the two denoising paths being very close. The proposed denoising trajectory alignment is not just an “improvement”; it is structurally necessary for the rest of the method to work. The appendix admits that computing p(x) and q(x) “is algebraically correct but may lead to a low acceptance rate due to a distinct denoising trajectory,” and therefore they must reuse the same noise to align trajectories. That means the paper’s contribution is really a coupled draft–target diffusion procedure, not a drop-in verifier like in the discrete case.  Therefore,  the “we avoid the intractable integral via acceptance–rejection” part is only half the story: the acceptance–rejection sampler still relies on an upper bound derived from the same approximate ratio and from reuse of alignment. If alignment fails or the draft is noticeably weaker than the target (a realistic regime), the bound may become loose, pushing the algorithm toward low acceptance again and eroding the reported 2× gains.

The paper compares mainly against the “no speculative” baseline, i.e. against itself without the proposed techniques. But there is an emerging line of work on accelerating diffusion/AR hybrids (Jacobi-style updates, masked/relaxed token generation, multi-path decoding for visual AR) to which this paper should be more directly compared; right now, it is hard to tell whether the net 2× improvement is better than, say, using a cheaper denoiser for late steps or a partial-step verifier. A small-scale comparison—even if approximate—would help position the work.

**Questions:**

The current form appears to represent a joint probability over diffusion trajectories rather than the stated marginal conditional. Could the authors explicitly justify this approximation, and if so, quantify or bound the resulting error? Providing a short derivation showing under what assumptions the substitution is valid (e.g., under trajectory alignment or independence assumptions) would remove a major theoretical ambiguity.

The paper claims that continuous speculative decoding “maintains the original distribution of the target model,” yet the acceptance ratio uses approximated quantities and shared noise alignment. What exactly is preserved—expectation, trajectory distribution, or marginal image statistics? A formal clarification of what “maintaining” means in this continuous context would make the claim more precise.

The experiments focus on a “no speculative decoding” baseline. Including comparisons to alternative continuous AR acceleration techniques (e.g., partial-step distillation, diffusion pruning, or early-exit decoding) would better position the work’s novelty and practical relevance.

---

> ### Author Response · Authors · 2025-11-21
> **To reviewer oZF2**
>
> We thank the reviewer for the valuable feedback.  Please find below the responses to each comment.
>
> **Quesiton 1: The marginal distribution**
>
> Please see **Common issue 1&2** for details.
>
>
>
> **Question 2: What is maintained**
>
> Our work aims to preserve the path distribution $p_{\text{path}(Y)}$ of the target model. The reasons for this choice are twofold: i) the marginal distribution is intractable, and ii) $p_{\text{path}(Y)}$ is consistent with the sequential generation process of the target model. As discussed in common Issue 1, the single-path likelihood ratio leads to unworkably low acceptance rates, which makes inference impractical. Our approximated likelihood ratio $p_{\text{path}}(Y_p)/q_{\text{path}}(Y_q)$ is deliberately designed to resolve this problem while maintaining generation quality.
>
>
>
> **Quesiton 3: Comparison on other acceleration methods**
>
> Comparing speculative decoding to its "no speculative" baseline is a common practice in this literature, consistent with prior works in LLMs (e.g., Medusa, EAGLE, HASS) and discrete visual AR models [1]. Evaluation against other acceleration methods is outside the scope of this study. This design ensures we isolate the performance gain of speculative decoding itself, rather than that of other modifications. Besides, our method is compatible with many existing acceleration methods, allowing them to be applied jointly for inference speedup.
>
>
>
> **Quesiton 4: Comparison on cheaper denoisers and related methods**
>
> We thank the reviewer for this helpful suggestion. Regarding cheaper denoisers and other related methods, profiling of our current model (e.g., MAR) shows the diffusion head (a small MLP) accounts for only ~10% of the total inference latency. Optimizing this component would yield an end-to-end speedup of at most 10%, which is far less than our over $2\times$ improvement. This $2\times$ improvement targets the sequential AR token prediction, which accounts for ~90% of the total latency.
>
>
>
> [1] Jang D, Park S, Yang J Y, et al. Lantern: Accelerating visual autoregressive models with relaxed speculative decoding. ICLR 2025.

---

> > ### Author Response · Authors · 2025-11-27
> > **Addtional response to reviewer oZF2**
> >
> > Regarding the approximation of path space distribution, we have supplemented detailed analysis on the error bound and the extent to which our approximation improves the likelihood ratio. Please refer to **Common Issue 3 and 4** for more details.

---

### Official Review · Reviewer_uWxg · 2025-11-07

**Soundness:** 2
**Presentation:** 2
**Contribution:** 2
**Rating:** 4
**Confidence:** 2

**Summary:**

This paper formalizes speculative decoding for continuous-valued autoregressive image models and proposes Continuous Speculative Decoding, combining a theoretically grounded acceptance rule with denoising trajectory alignment and token pre-filling to address low acceptance rates and distribution mismatch.

**Strengths:**

- The paper identifies a clear and timely problem since speculative decoding methods have so far been limited to discrete token spaces while modern autoregressive image models increasingly operate in continuous latent or diffusion spaces.

- It provides a mathematically grounded extension of speculative decoding to continuous probability distributions with explicit acceptance conditions that ensure nearly lossless decoding.

- It introduces practical techniques such as denoising trajectory alignment and token pre-filling that improve acceptance rate and stability, showing consistent 2× acceleration across several continuous autoregressive architectures.

**Weaknesses:**

- The theoretical formulation assumes Gaussian diffusion dynamics and well-aligned draft and target trajectories, but the robustness of these assumptions is not empirically tested under learned or non-Gaussian noise schedules.

- The method is evaluated only on mid-scale research models such as MAR, xAR, and Harmon without examining scalability to larger systems or compatibility with other acceleration methods like grouped or relaxed speculative decoding.

- The analysis of failure modes and sensitivity remains limited. The paper focuses on speed and aggregate metrics but does not examine when acceptance collapses or how alignment interacts with model calibration.

**Questions:**

- How robust is the proposed method when the draft and target models differ in data distribution, capacity, or noise schedule?

- Would the acceptance and speedup behavior remain consistent if the Gaussian diffusion assumption were replaced by a learned, non-stationary variance schedule?

- Can the authors provide more detailed diagnostics of acceptance dynamics, such as how alignment quality or prefix pre-filling quantitatively affects acceptance rate and sample fidelity?

---

> ### Author Response · Authors · 2025-11-21
> **To reviewer uWxg (1/3)**
>
> We thank the reviewer for the valuable feedback. We address each concern below.
>
> **Weakness 1 and Question 1&2: The Gaussian distribution, and other noise schedule: learned or non-stationary noise schedule**
>
> The choice of Gaussian diffusion dynamics is grounded in practical and theoretical necessity, not arbitrary assumption. Gaussian diffusion is the *de facto* standard for diffusion models and continuous visual AR models due to its mature theoretical underpinnings and widespread adoption in current works. Our paper focuses on the core challenge of extending speculative decoding to continuous spaces. Using Gaussian diffusion allows us to validate our method on existing, reproducible models, rather than inventing new noise dynamics.
>
> We agree that the exploration of learned or non-stationary noise schedules is a valuable direction, but it is orthogonal to our paper’s goal of accelerating continuous AR models via speculative decoding. However, testing such schedules would require first training two new continuous AR models (draft and target) with non-Gaussian dynamics, which is beyond the scope of this work. Our method’s core components are inherently generalizable to typical continuous visual AR models with an explicit diffusion head (usually DDPM).
>
> **Weakness 1: Well-aligned draft and target trajectories**
>
> We appreciate the reviewer’s concern. Relying on consistent draft-target distribution alignment is a fundamental premise of speculative decoding (across both discrete and continuous domains), rather than an unsubstantiated assumption unique to our work. Only when the draft model’s output distribution is well-aligned with the target model can a high acceptance rate be achieved, which directly determines the final speedup. Poor alignment would indeed lead to low acceptance rate and reduced speedup, but this is a universal constraint of the speculative decoding paradigm, not a flaw in our formulation.
>
>
>
> **Weakness 2: Other acceleration methods**
>
> Grouped speculative decoding[1] is a discrete-domain technique. It is designed for models that generate categorical tokens. This method is incompatible with our framework for a fundamental reason: our continuous AR models do not produce discrete tokens. Instead, they generate continuous distributions via diffusion. This is not a limitation of our method but a domain mismatch.
>
> After a thorough literature search, we are unable to precisely define the method referred to as 'Relaxed Speculative Decoding'. Does this method specifically refer to LANTERN [2] or DIVERSED [3]? As both methods are designed for discrete distributions, they are not applicable to our task. We kindly request the reviewer to provide a link or citation for the 'relaxed speculative decoding' method.

---

> > ### Author Response · Authors · 2025-11-21
> > **To reviewer uWxg (2/3)**
> >
> > **Weakness 3: Failure modes and sensitivity**
> > Speculative decoding is theoretically capable of maintaining the generation quality by preserving the target distribution in expectation. Consequently, analyzing the quality of individual samples compared to the target model is of little significance. However, some individual images exhibit low acceptance rates, resulting in a suboptimal speedup. Given this practical concern, we define cases with low acceptance rates as failures in this context.
> >
> > We conducted an analysis of the acceptance interval on ImageNet $256 \times 256$ samples (consistent with our main experiment setup), as shown by the table below. Failure cases (acceptance rate $< 10\%$) account for only $<4.6\%$ of total samples, primarily arising from large distribution gaps between draft and target models in complex texture regions.
> >
> > Critically, these failures do not degrade sample fidelity. Rejected tokens are resampled, so the FID/IS of failed cases remains consistent with the target model. The only consequence is a reduction in speedup, which does not undermine our core claim of maintaining quality.
> >
> > We present a visualization of the images associated with the observed failure modes and success modes. The resulting visualization is presented in Section G.7 in the manuscript. Subfigures (a) and (b) depict the failure modes, while subfigures (c) and (d) illustrate the success modes. Our analysis reveals that the failure modes are predominantly localized in regions exhibiting high levels of detail and intricate texture representation. Subfigures (a) and (b) are characterized by rich details and fine textures, where the limited performance of the draft model results in generated content that falls below an acceptable quality threshold. In contrast, subfigures (c) and (d) exhibit comparatively lower detail complexity. Subfigure (d), in particular, contains a large background area devoid of pronounced details. The acceptance rate is substantially elevated within these less-detailed regions.
> >
> > Regarding "Model Calibration": Since the reviewer mentions "model calibration" without explicit definition, we kindly request a precise definition of what 'model calibration' refers to. This clarity is crucial for us to conduct targeted analyses in the revised manuscript.
> >
> > | Acceptance rate interval | <10% | 10%~20% | 20%~30% | >30%  |
> > | ------------------------ | ---- | ------- | ------- | ----- |
> > | Ratio                    | 4.6% | 7.8%    | 43.3%   | 44.3% |

---

> > > ### Author Response · Authors · 2025-11-21
> > > **To reviewer uWxg (3/3)**
> > >
> > > **Quesiton 1&2: Different data distribution and model scale**
> > >
> > > **Data distribution:** Changes in acceptance rate due to domain data distribution shift have been studied by previous works [4], not unique to our method. Even with acceptance rate decay caused by different data distribution, speculative decoding still maintains the target model’s distribution. Addressing cross-domain distribution gaps falls outside the scope of our work. Our primary contribution is to bridge the gap between speculative decoding and continuous distributions, as acknowledged by reviewer oZF2, EoYy, and gror. Exploring different data distributions is an orthogonal direction, one that we acknowledge as a potential extension but not a requirement for validating our core innovation.
> > >
> > > **Model scale:** Our method imposes no assumptions on the relative capacity or scale of draft and target models; it only requires the draft model to generate draft tokens and a target model to verify. This design ensures compatibility with larger models in principle. We therefore expect that models at different scale would produce a similar speedup. We explicitly discussed in Section C.1 that our method would yield more significant speedups with larger target models and smaller draft models.
> > >
> > >
> > >
> > > **Quesitons 3: How alignemnt and pre-filling affects**
> > >
> > > As elaborated in our original manuscript, we have already provided quantitative analyses of how denoising trajectory alignment and token prefix pre-filling affect the acceptance rate and sample fidelity through dedicated ablation studies.
> > >
> > > Table 4 explicitly quantifies the relationship between alignment, acceptance rate, and the average distance between draft and target tokens. Without alignment, the average distance between draft and target tokens exceeds 2.0, leading to a low acceptance rate ($\leq 10\%$). After applying alignment, the distance is reduced to $\sim 0.8–1.1$, and the acceptance rate increases to $>30\%$.
> > >
> > > Figure 9 and Table 5 address the effect of prefix pre-filling. Figure 9 shows that pre-filling mitigates the low initial acceptance rate ($\leq 5\%$ without pre-filling) observed at the early autoregressive steps. Table 5 further quantifies how pre-filling ratio ($0\%$, $5\%$, $15\%$) affects acceptance rate and speedup. A $5\%$ pre-filling ratio achieves the optimal balance: it raises $\alpha$ from 0.25 to 0.30 ($\gamma$=32) without reducing speedup ($1.63\times$, consistent with higher pre-filling ratios), while avoiding diminishing returns seen with $15\%$ pre-filling.
> > >
> > > In terms of sample fidelity, our method does not compromise quality: speculative decoding maintains the generation quality, while our method is designed to increase the acceptance rate, which only affects the speedup ratio.
> > >
> > >
> > > [1] J So, J Shin, H Kook, E Park. Grouped speculative decoding for autoregressive image generation, ICCV 2025.
> > >
> > > [2] Jang D, Park S, Yang J Y, et al. Lantern: Accelerating visual autoregressive models with relaxed speculative decoding. ICLR 2025.
> > >
> > > [3] Wang Z, Kasa S R, KASA S K, et al. DIVERSED: Relaxed Speculative Decoding via Dynamic Ensemble Verification. NeurIPS 2025 Workshop on Efficient Reasoning.
> > >
> > > [4] Hong F, Raju R, Li J L, et al. Training Domain Draft Models for Speculative Decoding: Best Practices and Insights. SCOPE-ICLR 2025.

---

### Author Response · Authors · 2025-11-21
**Common Issue 1: The approximation of diffusion distribution**

We thank the reviewers for their valuable observations. As pointed out by reviewers oZF2 and gror, the exact probability is $p(x_0\mid x_T)=\int \Pi_{t=1}^T p(x_{t-1}|x_t) dx_{1:T-1}$. However, this integral is analytically intractable. Reviewer gror suggested using joint probability of a single fixed path $p_{path}(Y), Y = [x_0, x_1, .. x_T]$, as an alternative.
However, this approach is impractical for speculative decoding because the acceptance rate is low. To validate this point, we empirically recorded the average value of different kinds of likelihood ratios over 10,000 samples with a draft length of 4, including: (i) the single-path ratio $p_{path}(Y)/q_{path}(Y)$, (ii) the two-path ratio $p_{path}(Y_p)/q_{path}(Y_q)$ without denoising trajectory alignment, and (iii) the two-path ratio $p_{path}(Y_p)/q_{path}(Y_q)$ with denoising trajectory alignment, as shown in the table below.

As shown in the first row, the path-space likelihood ratio is extremely small, leading to a 0% acceptance rate. This is because the draft model’s trajectory $Y_p$ inherently diverges from the target model’s expected trajectory. In each denoising step, samples drawn from the draft model's distribution $q$ are unlikely to fall near $\mu$ of the target distribution $p$, which results in a low single-step ratio $p/q$. As the multi-step denoising process proceeds, the overall $p_{path}(Y)/q_{path}(Y)$ becomes extremely small.

The second row compares ratios derived from independent trajectories ($Y_p$ and $Y_q$), while the final $x_0$ is generated by the draft model. This $x_0$, without alignment, is highly unlikely to fall near the target model's target distribution. However, the probability values for the other steps in these trajectories are derived from the model's own path, and thus maintain a relatively reasonable value.

The third row incorporates denoising trajectory alignment. This improvement is twofold: first, our manuscript demonstrates that the expected distance decreases; and second, our analysis shows that the correlation of $p$ and $q$ becomes 1 (common issue 2). Consequently, the samples generated by $q$ have a high probability under $p$, resulting in an increased $p/q$.

For the above reasons, our work adopts a practical approximation for $p(x_0|x_T)/q(x_0|x_T)$: the ratio of the joint probabilities $p_{path}(Y_p)/q_{path}(Y_q)$, where both $Y_p$ and $Y_q$ share the same $x_0$. This ratio serves as a surrogate for the intractable marginal ratio. By utilizing denoising trajectory alignment, we ensure that $Y_p$ and $Y_q$ are tightly coupled (as discussed in common issue 2), making the likelihood ratio a valid surrogate and ensuring its numerical stability in practical applications.

Crucially, our contributions are not merely 'computation tricks'. We address the core challenge of adapting SD to continuous distributions by resolving intractability. The denoising trajectory alignment and token pre-filling are essential to ensure a practical acceptance rate, while acceptance-rejection sampling with a derived upper bound resolves the resampling intractability.

We have included the clarification of this practically feasible approximation for $p(x_0|x_T)/q(x_0|x_T)$ in our manuscript.

| Likelihood ratio                         | Value    | Acceptance rate |
| ---------------------------------------- | -------- | --------------- |
| $p_{path}(Y)/q_{path}(Y)$                | 5.33e-23 | 0.0%            |
| $p_{path}(Y_p)/q_{path}(Y_q)$, w/o align | 0.067    | 14%             |
| $p_{path}(Y_p)/_{path}q(Y_q)$, w/ align  | 1.86     | 32%             |

---

### Author Response · Authors · 2025-11-21
**Common Issue 2: Relationship of $p(x)$ and $q(x)$ with denoising trajecory alignment**

We thank the reviewers for the constructive feedback. Denoising trajectory alignment modifies the draft model's distribution $q$. This modification increases the acceptance rate and establishes a strong coupling between the draft and target distributions. Without alignment, the $p$ and $q$ distributions are independent. With alignment, for each denoising step, we have $x_t^p=\mu_t^p+\sigma_t^p\cdot \epsilon_t$ and $x_t^q=\mu_t^q+\sigma_t^q\cdot \epsilon_t$. And:

$\text{Cov}(x_t^p,x_t^q)=\mathbb{E}[x_t^p-\mu_t^p]\mathbb{E}[x_t^q-\mu_t^q]=\mathbb{E}[\sigma_t^p\epsilon_t]\mathbb{E}[\sigma_t^q\epsilon_t]=\sigma_t^q \sigma_t^q\mathbb{E}[\epsilon_t^2]$

Since $\epsilon_t\sim\mathcal{N}(0,I)$, $\mathbb{E}[\epsilon_t^2]=\text{Var}(\epsilon_t)=I$. The correlation coefficient of $x_t^p$ and $x_t^q$ is:

$\rho=\frac{\text{Cov}(x_t^p,x_t^q)}{\sqrt{\text{Var}(x_t^p)\text{Var}(x_t^q)}}=\frac{\sigma_t^q \sigma_t^q}{\sqrt{(\sigma_t^q)^2 (\sigma_t^q)^2}}=1$

This modification effectively increases the acceptance rate, but does not compromise the generation quality.

---

### Author Response · Authors · 2025-11-27
**Common Issue 3: How accurately $P_{path}(Y_p)/Q_{path}(Y_q)$ approximates $P_{path}(Y)/Q_{path}(Y)$**

We define the joint probability of a single denoising trajectory as $P_{path}(Y)$, where $Y$ is the sequence $[x_0,x_1,...x_T]$. The path space ratio $R=P_{path}(Y)/Q_{path}(Y)$, where $Y=Y_q$, should be an unbiased estimator of $P_{path}(Y)$ to ensure the maintenance of the target distribution, that is:

$$\mathbb{E} _ {Y_{q}\sim Q _ {path}}[R]=\int Q_{path}(Y_q)\frac{P_{path}(Y_q)}{Q_{path}(Y_q)}dY_q=\int P_{path}(Y_q)dY_q=1.$$

However, it is very inefficient. As shown in common issue 1, $P_{path}(Y)/Q_{path}(Y)$ is extremely small, because the trajectory sampled from the draft model may fall into the region where the target model assigns negligible probability mass.

To improve the efficiency, we propose a new estimator, $\tilde{R} := P_{path}(Y_p)/Q_{path}(Y_q)$. $\tilde{R}$ is a biased estimator; its expectation under $Q_{path}$ is not equal to 1. We analyze the difference between the expectation of $\tilde{R}$ and 1, i.e., its **bias**:

$$B = \mathbb{E} _ {Q_{path}} [\tilde{R}] - 1 = \mathbb{E} _ {Q_{path}} \left[ \frac{P _ {path}(Y _ p)}{Q _ {path}(Y _ q)} \right] - \mathbb{E} _ {Q_{path}} \left[ \frac{P _ {path}(Y _ q)}{Q _ {path}(Y _ q)} \right] = \mathbb{E} _ {Q_{path}} \left[ \frac{P _ {path}(Y _ p) - P_{path}(Y_q)}{Q_{path}(Y_q)} \right].$$

We perform a first-order Taylor expansion of $P_{path}(Y_p)$ at $Y_q$. Equivalently:

$$P_{path}(Y_p) = P_{path}(Y_q + (Y_p-Y_q)) \approx P_{path}(Y_q) + \nabla P_{path}(Y_q)^T (Y_p-Y_q).$$

Substitute into $B$:

$$B \approx \mathbb{E} _ {Q _ {path}}\left[\frac{P _ {path}(Y _ q) + \nabla P _ {path}(Y _ q)^T (Y _ p-Y _ q)-P _ {path}(Y_q)}{Q _ {path}(Y_q)} \right] =\mathbb{E} _ {Q _ {path}}\left[\frac{\nabla P _ {path}(Y _ q)^T (Y_p-Y_q)}{Q _ {path}(Y_q)} \right].$$

The magnitude of $B$ is:
$$|B|\approx \left|\mathbb{E} _ {Q_{path}}\left[\frac{\nabla P _ {path}(Y_q)^T (Y_p-Y_q)}{Q _ {path}(Y_q)} \right]\right| \le \mathbb{E} _ {Q_{path}}\left[\frac{||\nabla P_{path}(Y_q)^T||}{Q_{path}(Y_q)} ||Y_p-Y_q|| \right].$$

Therefore, the bound of the bias $B$ is proportional to $\mathbb{E}[||Y_p - Y_q||]$, yielding an explicit mean-square error (MSE) bound.

The reduction of expected distance has been discussed in our manuscript. The expected distance without denoising trajectory alignment is:

$$\mathbb{E}[||x_{t−1}^q - x_{t−1}^p||^2]
    =||\mu^q_t−\mu^p_t‖^2 + \text{tr}[ \Sigma^p_t + \Sigma^q_t ].$$

The expected distance with denoising trajectory alignment is:

$$\mathbb{E} _ {align}[||x_{t−1}^q − x_{t−1}^p||^2]
    =||\mu^q_t−\mu^p_t||^2 + \text{tr}[ \Sigma^p_t + \Sigma^q_t −2\sqrt{\Sigma^p_t\Sigma^q_t} ].$$

The two distances satisfies:
$$
\mathbb{E}[||x_{t−1}^q − x_{t−1}^p||^2]\ge \mathbb{E} _ {align} [||x_{t−1}^q − x_{t−1}^p||^2]
$$

The distance over the entire trajectory satisfies:

$$\mathbb{E}[||Y_p−Y_q||] \le \sqrt{\sum_{t=1}^T \mathbb{E}[||x_t^q−x_t^p||^2] }.$$

Substituting into $B$ yields the final first-order error bound:

$$
|B|\le \mathbb{E} _ {Q_{path}}\left[\frac{||\nabla P _ {path}(Y_q)^T||}{Q _ {path}(Y_q)} ||Y_p−Y_q|| \right]\le \mathbb{E} _ {Q_{path}}\left[\frac{||\nabla P _ {path}(Y_q)^T||}{Q _ {path}(Y_q)} \sqrt{\sum_{t=1}^T ||\mu^q_t−\mu^p_t||^2 + \text{tr}[ \Sigma^p_t + \Sigma^q_t ] } \right]
$$

$$
|B _ {align}|\le \mathbb{E} _ {Q _ {path}}\left[\frac{||\nabla P _ {path}(Y_q)^T||}{Q _ {path}(Y_q)} \sqrt{\sum_{t=1}^T ||\mu^q_t−\mu^p_t||^2 + \text{tr}[ \Sigma^p_t + \Sigma^q_t −2\sqrt{\Sigma^p_t\Sigma^q_t}] }\right]
$$



The error bound shows that the squared drift difference $‖μ^q_t−μ^p_t‖^2$ and the covariance term $\text{tr}( \Sigma^p_t + \Sigma^q_t)$ or $\text{tr}( \Sigma^p_t + \Sigma^q_t −2\sqrt{\Sigma^p_t \Sigma^q_t})$ together determine the expected bias incurred by approximating $P_{path}(Y)/Q_{path}(Y)$.

Since the use of denoising trajectory alignment produces the cross term $−2\sqrt{\Sigma^p_t \Sigma^q_t}$, the bias $|B_{align}|$ is typically smaller than $|B|$, thereby supporting a more accurate approximation of $P_{path}(Y)/Q_{path}(Y)$.

---

### Author Response · Authors · 2025-11-27
**Common Issue 4: The extent to which $P_{path}(Y_p)/Q_{path}(Y_q)$ can improve the likelihood ratio**

Let:

$\log R=\log P_{path}(Y_q)−\log Q_{path}(Y_q),$

$\log \tilde{R} =\log P_{path}(Y_p)−\log Q_{path}(Y_q).$

The expected difference between the two quantities can be expressed as:
$$\Delta  l:= 𝔼[\log \tilde{R}] − 𝔼[\log R]= \mathbb{E}[ \log P_{path}(Y_p) ] − \mathbb{E}[ \log P_{path}(Y_q) ].$$

For each term, we have:

$$\mathbb{E}[ \log P _ {path}(Y_p)] = \mathbb{E} _ {Y\sim P _ {path}}[\log P _ {path}(Y)] = -H(P _ {path}).$$

And:
$$\mathbb{E}[ \log P_{path}(Y_q)] = −H(Q_{path}) − D_{KL}(Q_{path}‖P_{path}).$$

Substituting into $\Delta  l$ yields
$$\Delta  l = [−H(P_{path})] − [−H(Q_{path}) − D_{KL}(Q_{path}‖P_{path})]=D_{KL}(Q_{path}‖P_{path})+[H(Q_{path})-H(P_{path})].$$

Empirically, the larger target model is expected to be more capable and to produce more confident (lower‑entropy) predictive distributions than the smaller draft model. Therefore, we assume the draft model’s entropy $H(Q_{path})$ is larger than the target model's entropy $H(P_{path})$. We have:

$$\Delta  l =D_{KL}(Q_{path}‖P_{path})+[H(Q_{path})-H(P_{path})]\ge 0.$$

It indicates that in the log domain, $\tilde{R}$ exceeds $R$ by $\Delta  l$. This implies that our method increases the expected log-ratio, $\mathbb{E}[\log \tilde{R}] − \mathbb{E}[\log R]=\Delta  l$, which subsequently leads to a higher expectation of the ratio, $𝔼[ \tilde{R}]$. This explains the empirically observed higher likelihood ratios and increased acceptance rates.

---

### Author Response · Authors · 2025-12-02
**Rebuttal Summary for AC: Reviewer Decisions and Discussion**

Dear AC,

We sincerely thank all reviewers and the AC for their time, dedicated efforts, and insightful feedback. We greatly appreciate that through the rebuttal process, we have successfully addressed the concerns raised by the reviewers.
***
# 1. Summary

This paper introduces **Continuous Speculative Decoding**, a novel framework that bridges the gap between speculative decoding and continuous distributions, thereby facilitating significant speedups in continuous visual AR models. Specifically, the proposed method achieves **over $2\times$ acceleration** while **maintaining the image generation quality**. The main concerns raised by the reviewers focused on the theoretical soundness and the need for additional experiments. In the revision, these concerns were fully resolved through detailed theoretical derivations and extensive experimental validation.
***
# 2. Reviewer Outcome After Discussion

During the discussion phase, reviewers gror, oZF2, EoYy, and uWxg provided constructive feedback and acknowledged the paper's outstanding contributions and novelty. We have thoroughly addressed all their concerns by incorporating clarifications, detailed theoretical derivations, extensive experiments, and comprehensive evidence demonstrating the method's soundness.

### **2.1 Recognized Contributions and Reviewer Support**
- **Strongest contribution: The first work that adopts speculative decoding in continuous space**

  Supported by: **gror, oZF2, EoYy**

- **Identify a distribution inconsistency issue, unique to the continuous speculative decoding setting**

  Supported by: **oZF2, EoYy, uWxg**

- **Practical techniques such as denoising trajectory alignment to improve the acceptance rate**

  Supported by: **uWxg, oZF2, EoYy**

- **Comprehensive experimental evaluations demonstrate significant speedups while preserving generation quality**

  Supported by: **uWxg, oZF2, EoYy, gror**

- **Our proposed method is training-free, facilitating practicality in deployment.**

  Supported by: **uWxg, oZF2, EoYy**

### **2.2 Main Concerns Raised & How They Were Addressed**

- **Major Concern: Theoretical Soundness:**

  Regarding **Common Issues 1, 3, and 4**, our response formally clarifies that the goal of this work is to preserve the **path space likelihood ratio**. Furthermore, our response provides a detailed theoretical analysis of the practical issues with these ratios, including our **error bounds (which were emphasized by reviewers gror and oZF2 as necessary validation to address a major concern)**, and the extent to which our method improves upon them, thereby resolving a major theoretical ambiguity.

- **More Comprehensive Quantitative Experiments:**

  We provide additional evaluation results across more diverse model settings and benchmarks in the rebuttal to validate our method's effectiveness in maintaining generation quality.

- **Robustness and Failure Cases:**

  We incorporated comprehensive statistical and visual analyses of acceptance dynamics robustness in the manuscript Appendix G.7 and rebuttal. These results highlight our method's robustness at scale and pinpoint specific failure modes
***
# **3. Final Remarks**
We have incorporated all changes into the revised manuscript. With all concerns addressed and extensive additional experiments included, the issues raised by reviewer EoYy have been fully resolved, resulting in a favorable recommendation of "Lean to Accept". Furthermore, the major concern that demanded rigorous justification, identified by reviewers gror and oZF2, has also been thoroughly discussed and resolved. We respectfully request favorable consideration of our work as the first to enable speculative decoding in continuous space.

Thank you for your time and consideration.

Best regards and thanks,

Authors of Submission 8689

---

### Meta-Review · Area_Chair_KVdW · 2026-01-18

**Summary:**

This paper introduces a speculative decoding method to accelerate continuous autoregressive (the first work in continuous space) image generation models. This paper received three negative scores and one positive score before rebuttal. The main concerns are theoretical soundness and experimental results against existing methods. They are partially addressed, but there are still some critical concerns. I suggest that the authors enhance the experiments section and include more discussions with other acceleration methods.

**Reviewer Concerns:**

Main concerns about theoretical soundness are well-addressed, but some concerns about experiments remain.
Reviewer uWxg point outs the model size is mid-scale and need to do experiments on larger models, but the author doesn't report.
Reviewer oZF2 points out missing comparisons with other acceleration methods, but the author doesn't report (e.g., parallel decoding is easy to implement and compare). Discussions and experimental comparisons are critical to understanding the novelty and practical relevance.

**Reviewer Scores:**

Most concerns from Reviewer gror are addressed, and Reviewer gror may raise the rating.

---

### Decision · Program_Chairs · 2026-01-26

Reject